# Environmental pH and peptide signaling control virulence of *Streptococcus pyogenes* via a quorum-sensing pathway

Hackwon Do[1,2], Nishanth Makthal[1,2], Arica R. VanderWal[1,2], Matthew Ojeda Saavedra[1,2], Randall J. Olsen[1,2,3], James M. Musser[1,2,3] & Muthiah Kumaraswami [1,2]

Bacteria control gene expression in concert with their population density by a process called quorum sensing, which is modulated by bacterial chemical signals and environmental factors. In the human pathogen *Streptococcus pyogenes*, production of secreted virulence factor SpeB is controlled by a quorum-sensing pathway and environmental pH. The quorum-sensing pathway consists of a secreted leaderless peptide signal (SIP), and its cognate receptor RopB. Here, we report that the SIP quorum-sensing pathway has a pH-sensing mechanism operative through a pH-sensitive histidine switch located at the base of the SIP-binding pocket of RopB. Environmental acidification induces protonation of His144 and reorganization of hydrogen bonding networks in RopB, which facilitates SIP recognition. The convergence of two disparate signals in the SIP signaling pathway results in induction of SpeB production and increased bacterial virulence. Our findings provide a model for investigating analogous crosstalk in other microorganisms.

[1] Center for Molecular and Translational Human Infectious Diseases Research, Houston Methodist Research Institute, Houston, TX 77030, USA.
[2] Department of Pathology and Genomic Medicine, Houston Methodist Hospital, Houston, TX 77030, USA. [3] Department of Pathology and Laboratory Medicine, Weill Medical College of Cornell University, New York, NY 10021, USA. Correspondence and requests for materials should be addressed to M.K. (email: mkumaraswami@houstonmethodist.org)

Bacterial pathogens survive in complex milieus in the host. They encounter diverse arrays of host-derived innate defense mechanisms including environmental alterations, oxidative stress, nutrient limitation, and immunologic factors[1–4]. As a countermeasure, successful pathogens have sophisticated signaling pathways to sense their immediate environment and orchestrate appropriate transcriptional responses that mediate adaptation in vivo[5,6]. Quorum sensing pathways monitor alterations in bacterial population density and control expression of genes involved in crucial cellular processes including virulence[7–9]. Quorum sensing involves signal secretion, signal recognition by specific receptors in the neighboring cells, and transcription regulation of the target genes by signal-bound receptors[7–9]. In addition to endogenous bacterial signals, quorum-sensing gene networks (regulons) are also controlled by environmental factors such as pH[10–17]. Bacterial quorum sensing and pH-sensing mechanisms have been studied extensively, but largely as two separate and unrelated gene regulatory processes[7–9,18–21]. As a consequence, relatively little is understood about the interplay between the two major bacterial signaling pathways. Importantly, the molecular mechanisms by which bacteria monitor environmental pH alterations and couple the signal perception to control quorum-sensing pathways remain unknown.

Group A streptococcus (GAS), also known as Streptococcus pyogenes, is a human-specific pathogen that causes a broad spectrum of diseases ranging from mild pharyngitis and impetigo to the life-threatening necrotizing fasciitis and streptococcal toxic shock syndrome[22]. Globally, GAS causes an estimated 616 million cases of pharyngitis, and 660,000 invasive infections that result in 163,000 deaths annually[23]. GAS produces several bacterial surface-associated and secreted virulence factors including a secreted cysteine protease, known as streptococcal pyrogenic exotoxin B (SpeB)[24]. SpeB is one of the most extensively characterized GAS virulence factor for its role in disease pathogenesis[25–32]. SpeB is produced during human infection and crucial for GAS virulence in several animal models of infection[26–28,33–35]. Proteolytic cleavage of host and bacterial proteins by SpeB contributes significantly to host tissue damage and disease dissemination[25]. Consistent with its contribution to pathogenesis, GAS employs elaborate transcriptional and post-transcriptional regulatory mechanisms to control spatiotemporal production of SpeB[25,30,31,35–39].

A noncanonical quorum-sensing pathway controls speB transcription[30,37,40,41]. The global gene regulator known as regulator of proteinase B (RopB) and an eight amino acid leaderless peptide signal, SpeB-inducing peptide (SIP), form an intracellular receptor and intercellular peptide signal pair that controls speB expression[37]. The SIP peptide is produced and secreted during high-bacterial population density and reimported into the bacterial cytosol, where it directly interacts with cytosolic RopB[37] (Supplementary Fig. 1a). SIP promotes high-affinity RopB-speB promoter interactions and RopB oligomerization. Subsequently, the oligomeric RopB bound to the speB promoter induces robust speB expression, a process that is operative during experimental mouse infection[37,42] (Supplementary Fig. 1a). Each component of the SIP signaling pathway must be functional for a wild-type virulence phenotype[35,37,43]. Thus, the SIP regulatory circuit is the primary signaling mechanism controlling speB transcription in vitro and in vivo.

It has been known for several decades that extracellular SpeB protease production occurs under acidic growth conditions in vitro[36,44–46]. Auto-acidification of the environment and speB expression occurs during high-GAS population density. However, it remains unclear whether environmental acidification is a physiological signal controlling SpeB biogenesis or an unrelated event occurring contemporaneously with speB expression.

Importantly, the signaling pathway(s) responsible for environmental pH sensing by GAS and molecular mechanism by which two seemingly disparate signals, environmental acidification and population density, converge to control quorum sensing-dependent SpeB production remain elusive.

In this report, we show that environmental acidification is the critical physiological signal that upregulates SpeB production by controlling speB transcription. GAS integrates a pH-sensing mechanism with the SIP signaling pathway through a pH-sensitive histidine switch in RopB. The protonated side chain of RopB histidine-144 in low-environmental pH likely engages in intramolecular hydrogen-bonding interaction with the neighboring Y176, Y182, and E185 located at the base of SIP-binding pocket. The putative pH-dependent allostery in RopB activates speB expression by promoting high-affinity binding of SIP to RopB. Given the pH dependence of quorum-sensing pathways in many bacteria[11–15], we propose that coupling the pH sensitivity of a histidine switch to signal recognition and gene regulation is a general allosteric strategy employed by quorum sensing regulators.

## Results

**Environmental Acidification Activates *speB* Expression.** Previous studies have demonstrated that SpeB protease production occurs in growth medium at low pH[36,44–46]. However, whether the environmental pH-mediated control of SpeB biogenesis is regulated at the transcriptional or post-transcriptional level is not fully understood. Thus, we tested the hypothesis that environmental pH controls speB expression by correlating GAS growth kinetics in laboratory medium (THY) to environmental pH changes and speB expression. Due to the fermentative metabolism of GAS, the pH of the growth medium gradually decreased from relatively neutral ($t = 0$ h, pH 7.4) to slightly acidic pH values ($t = 9$ h, pH 5.5) in concert with increasing GAS population density (Fig. 1a and Supplementary Fig. 2a, b). The onset of growth medium acidification (pH 5.5) coincided with induction of speB expression ($t = 9$ h, 3222-fold induction) (Fig. 1a), highlighting the link between acidic pH and speB expression. These observations led us to hypothesize that environmental acidification activates speB expression. To test this hypothesis, GAS was grown to the late-exponential phase (LE, $A_{600}$ ~1.5), swapped with THY medium adjusted to a different pH, and speB transcript level was assessed by quantitative real time polymerase chain reaction (qRT-PCR). The pH alterations had no effect on bacterial viability (Supplementary Fig. 2c). However, after 1 h incubation, GAS had maximal speB transcript level under acidic environmental pH (pH 5.5) (Fig. 1b). The transcript level of speB decreased in concert with the increments in pH, and speB expression was abolished at near or above neutral pH (Fig. 1b). Together, these data indicate that environmental acidification causes drastic induction of speB expression.

**Environmental pH controls *speB* expression during infection.** Given that environmental acidification activates speB expression in vitro, similar acidification of infected tissue and upregulation of speB expression may occur during infection. Thus, we hypothesized that infection with GAS inoculum prepared under pH conditions permissive for speB expression leads to early onset of lesion development. To test this hypothesis, we assessed the virulence of a GAS inoculum prepared in SpeB-producing (pH 6) or nonproducing pH (pH 8) conditions in a mouse model of necrotizing myositis. Consistent with our hypothesis, the low-pH inoculum caused rapid abscess development as early as 24 h postinoculation compared to high-pH inoculum (Fig. 1c, d). In contrast, no pH-dependent alterations in lesion character were

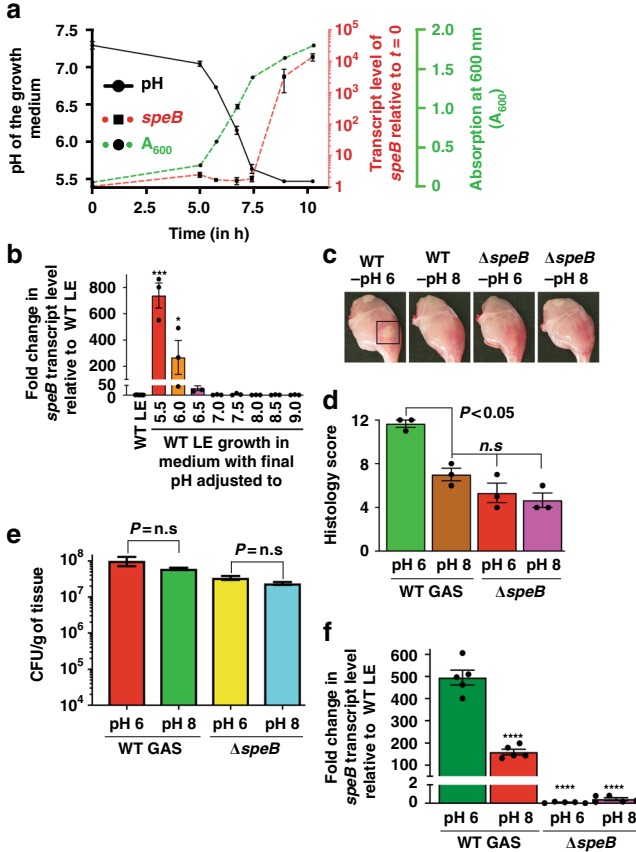

**Fig. 1** Environmental acidification controls *speB* expression. **a** Wild-type (WT) GAS was grown in THY broth, samples were collected at the indicated time points, and growth medium pH, *speB* transcript levels, and absorption at wavelength 600 nm ($A_{600}$) were determined. Right Y-axes represent fold-change in *speB* transcript levels (red) and $A_{600}$ (green). Fold-changes in transcript levels at indicated time points relative to starting culture (time point $t = 0$ h) are shown. Data are mean + standard deviation for three biological replicates. **b** WT GAS was grown in THY to late-exponential growth phase (LE, $A_{600}$ ~1.5), harvested by centrifugation, suspended in fresh THY adjusted to indicated pH and incubated for 1 h. The fold-change in *speB* transcript levels relative to WT-LE growth is shown. *P* values (*$P < 0.5$, ***$P < 0.001$) of the indicated samples relative to WT LE growth are shown. **c** Gross analyses of hindlimb lesions collected at 24 h postinfection from mice infected with $1 \times 10^7$ CFUs of each indicated strain. Larger lesion with extensive tissue damage in WT-infected mice in pH 6 is boxed (black box). **d** Histopathology scores of mouse muscle tissue infected with each indicated strain ($n = 3$ per strain). Data are mean + standard deviation. *P* values (*n.s.* = not significant) of the indicated strains were compared to WT GAS in pH 8. **e** Twenty mice were infected intramuscularly and mean colony-forming units (CFUs) recovered from the infected muscle tissue are shown. *n.s* indicates no statistical significance ($P > 0.05$). Data graphed are mean ± standard deviation. **f** Analysis of the *speB* transcript level in the intramuscular lesions from mice infected with indicated strains. Samples were collected at 24 h postinoculation from the lesions ($n = 4$ per strain) and analyzed in triplicate by qRT-PCR. The *speB* transcript levels in WT-LE ($A_{600}$ ~1.5) was used as a reference and fold-changes in *speB* transcript levels relative to the reference are shown. *P* values (****$P < 0.0001$) of the indicated strains were compared to WT GAS in pH 6. *P* values were determined by *t* test

observed in mice infected with an isogenic *ΔspeB* mutant strain (Fig. 1c, d). Importantly, mice infected with WT GAS in SpeB-producing (pH 6) or nonproducing pH (pH 8) conditions had similar bacterial burden (Fig. 1e), indicating that the observed

differences in lesion development between different pH groups are not due to reduced bacterial survival in vivo. However, compared to the WT strain grown to LE phase in THY, lesions from mice infected with low pH GAS inoculum (pH 6) had significantly higher *speB* expression than the inoculum in SpeB non-producing pH (pH 8) (Fig. 1f), suggesting that increased SpeB production in inoculum in SpeB-producing pH contributes to early onset of lesion development. Together, these results indicate that environmental acidification is a physiological signal in vivo that contributes to GAS pathogenesis.

**SpeB protease activity is pH dependent**. The pH dependency of *speB* expression led us to hypothesize that GAS produces SpeB under acidic pH conditions because SpeB auto-activation and protease activities are maximal in acidified environment. To test this hypothesis, we first assessed the effect of pH on SpeB auto-activation. SpeB is produced as an inactive zymogen ($SpeB_Z$, ~40 kDa) that subsequently undergoes autocatalysis to generate a mature active cysteine protease ($SpeB_M$, ~28 kDa). The protease activity of SpeB during recombinant protein overexpression and purification hampered the ability to purify $SpeB_Z$ to near homogeneity (Supplementary Fig. 3a, b). Nevertheless, the maturation process occurred more rapidly under slightly acidic conditions (pH 5.5 and 6.5) compared to near or above neutral pH (pH 7.5 and 8.5) (Supplementary Fig. 3a, b).

Next, we tested the pH dependence of $SpeB_M$ protease activity using the zymogen form of enzymatically inactive SpeB mutant (C192S) as a substrate[47]. Processing of $SpeB_Z$-C192S to mature form by $SpeB_M$ occurred rapidly at pH values between 5.5 and 6.5, whereas the enzymatic activity was drastically reduced at pH 8.5 (Supplementary Fig. 3c). Proteolytic cleavage by $SpeB_M$ was inhibited by addition of the cysteine protease inhibitor E64, indicating that the pH dependence of substrate cleavage is not due to pH-induced substrate instability and/or auto-degradation, but caused by the cysteine protease function of $SpeB_M$ (Supplementary Fig. 3c). Collectively, these results suggest that auto-activation and substrate cleavage activities of SpeB are maximal under slightly acidic environmental pH.

**Environmental pH controls *speB* expression via SIP signaling pathway**. The SIP signaling pathway is the primary regulatory mechanism controlling *speB* expression[37]. Thus, we hypothesized that SIP signaling functions at an optimal level in an acidified environment to activate *speB* expression. To test this hypothesis, we used a mutant strain designated *SIP** that is unable to produce endogenous SIP[37]. Thus, *speB* expression in the *SIP** mutant strain is dependent on exogenously added synthetic SIP peptide[37]. The *SIP** mutant strain grown to early stationary phase ($A_{600}$ ~1.7) was harvested and suspended in THY broth adjusted to different pH and supplemented with synthetic SIP. SIP-dependent *speB* expression was maximal within a very narrow pH range (pH 5.5–6.0) (Fig. 2a). The ability of SIP to induce *speB* expression decreased in concert with pH increase from neutral to basic pH (Fig. 2a). SIP lost the ability to induce *speB* expression at pH 8, even when SIP was added at 100-fold excess (Fig. 2a). The alterations in environmental pH alone were not sufficient to restore *speB* expression in the *SIP** mutant strain (Supplementary Fig. 4a). Similarly, SIP was functional only within a narrow range of acidic pH values (Fig. 2a), suggesting that both SIP and environmental pH act as co-dependent signals and control population density-dependent *speB* expression.

Exogenously added SIP had pH-dependent regulatory activity. Thus, we hypothesized that acidic pH activates *speB* expression by influencing one or more steps of the SIP signaling pathway. To test this hypothesis, we considered two possibilities: acidic

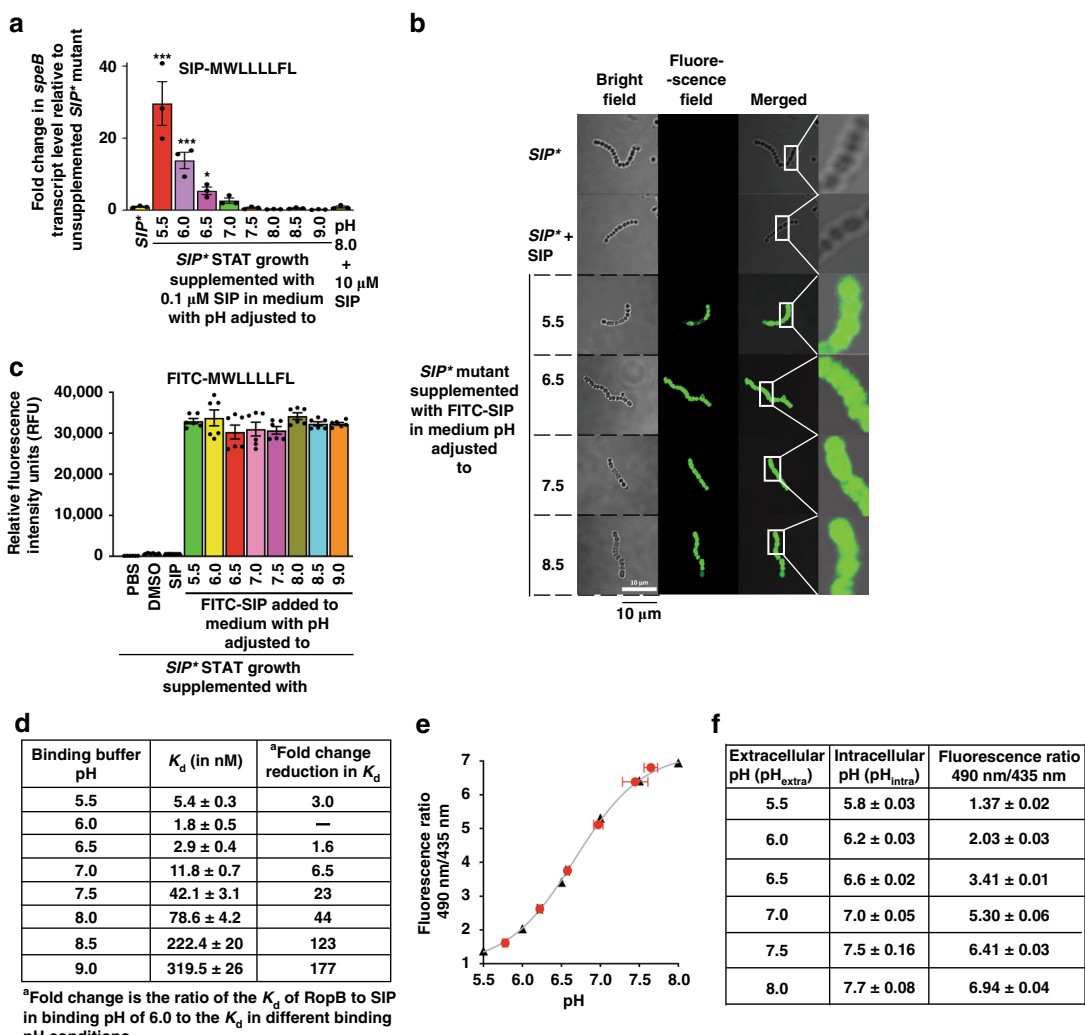

**Fig. 2** Environmental pH controls *speB* expression via SIP signaling pathway. **a** The *SIP\** mutant strain was grown to early stationary phase (STAT, $A_{600}$ ~1.7) and harvested by centrifugation. Bacteria were suspended in THY broth adjusted to indicated pH, supplemented with synthetic SIP and incubated for 1 h. The fold-changes in *speB* transcript levels relative to the unsupplemented *SIP\** mutant strain are shown. *P* values (\*\*\* *P* < 0.001) of the indicated samples were compared to unsupplemented GAS growth. *P* values were determined by *t* test. The amino acid sequence of the synthetic peptide SIP is shown in the inset. **b** Confocal microscopy images of isogenic *SIP\** mutant strain either unsupplemented or supplemented with indicated synthetic peptides in medium adjusted to indicated pH. For each sample, bright field, fluorescence field, merged images, and magnified views are shown. **c** The *SIP\** mutant strain was grown to STAT phase ($A_{600}$ ~1.7). Cells were transferred to THY broth adjusted to indicated pH and supplemented with either the indicated synthetic peptide or the carrier for the synthetic peptides (DMSO). After 1 h incubation, fluorescence measurements were obtained from clarified cell lysates using excitation and emission wavelengths of 480 and 520 nm, respectively. The changes in relative fluorescence units relative to the unsupplemented isogenic *SIP\** mutant strain are shown. The amino acid sequence of the synthetic peptide SIP with fluorescein modification at its amino-terminus (FITC-SIP) is shown in the inset. **d** RopB–SIP-binding constants assessed in binding buffer adjusted to indicated pH. **e** The relationship between pH and the ratio of cFSE intensities at wavelength 490 to 438 nm. The calibration curve with observed fluorescence ratio between fluorescence intensities at wavelength 490 to 435 nm in buffers adjusted to indicated pH (black triangles) is shown. The ratio of fluorescence intensities at wavelength 490 to 435 nm for cFSE-loaded GAS incubated in buffers adjusted to indicated pH are marked on the calibration curve (red circles). **f** The calculated GAS intracellular pH values in each tested extracellular pH values as determined by the equation derived from the calibration curve

environmental pH promotes (i) SIP reimport into the cytosol and/or (ii) SIP–RopB interactions. To investigate the pH dependency of SIP reimport, we used synthetic SIP peptide containing fluorescein modification at its amino-terminus (FITC-SIP) (Fig. 2b, c). FITC-SIP has regulatory activity comparable to unmodified SIP[37]. The *SIP\** mutant strain was incubated with FITC-SIP under different pH adjusted growth conditions and the cytosolic presence of FITC-SIP was assessed by fluorescence measurements and confocal microscopy. No significant differences in cytosolic fluorescence were observed among different pH adjusted growth conditions (Fig. 2b, c), suggesting that the

environmental pH alterations do not affect SIP import into the cytoplasm.

We next assessed the pH dependence of RopB–SIP interactions by a fluorescent polarization (FP) assay. We found that RopB and SIP interactions are pH-sensitive (Fig. 2d, Supplementary Fig. 4b). RopB engaged in high-affinity interaction with SIP under acidic pH conditions (pH 5.5–6.5), whereas RopB–SIP interactions were weaker at neutral or basic pH values (Fig. 2d, Supplementary Fig. 4b). These results suggest that the pH dependence of *speB* expression is due to the influence of pH on the association between RopB and SIP.

**GAS cytosol is acidified during environmental acidification.** The pH dependence of intracellular RopB–SIP interactions and RopB-dependent *speB* expression requires that GAS cytosol is acidified during environmental acidification. Thus, we determined GAS intracellular pH in response to environmental pH alterations. Using the pH-dependent fluorescence of fluorophore 6-carboxyfluorescein succinimidyl ester (cFSE)[48–50], we measured the intracellular pH of GAS incubated in different environmental pH. GAS grown to exponential phase ($A_{600}$ ~0.5) was incubated with the cell permeant cFSE precursor, the diacetate form of cFSE (cFDASE). The cFDASE is hydrolyzed to cFSE in the cytosol. The cytosolic fluorescent cFSE forms stable conjugate with the intracellular proteins, which prevents its leakage from the cells[48–50]. The cFSE-loaded cells were washed and suspended in different buffers adjusted for indicated pH. During GAS growth in vitro, pH of the growth medium decreases from 7.4 during early exponential phase of growth to 5.5 during stationary phase of growth (Fig. 1a). Thus, we have chosen an extracellular pH range of 5.5–8.0 in the pH measurement studies. After brief incubation in the indicated pH conditions, fluorescence intensities were measured. The relative ratios of fluorescence intensities between pH-sensitive (490 nm) and pH-insensitive (435 nm) excitation wavelengths were used to determine GAS intracellular pH.

Our results demonstrated that GAS intracellular pH decreases in response to environmental acidification. When GAS grown in neutral or above neutral pH, the extracellular and GAS cytosolic pH remained similar (Fig. 2e, f). However, in below neutral pH conditions, the intracellular pH decreased, and GAS maintain a pH difference ($\Delta pH = pH_{intracellular} - pH_{extracellular}$) of 0.2–0.3 units (Fig. 2e, f). Together, these results indicate that when environmental pH decreases to 5.5, as observed during high GAS population density (Fig. 1a), the intracellular pH (pH ~5.8) becomes conducive for optimal functioning of SIP signaling pathway.

**Structural basis of SIP recognition by RopB.** Our previous studies demonstrated that intact SIP in its native order of amino acid sequence is required for recognition by RopB[37]. However, the structural and biochemical basis for the interactions between RopB and SIP that dictate pH dependent and sequence-specific recognition of SIP by RopB, and the contribution of amino acids participating in RopB–SIP interactions to the regulation of *speB* expression remain unknown. Thus, to help elucidate the molecular basis for pH dependent, and sequence-specific SIP recognition by RopB, we crystallized the C-terminal domain of RopB (RopB–CTD) bound to SIP. Full-length RopB forms higher order oligomer upon SIP binding, which makes it less amenable for crystallization studies[37,40]. Thus, we used RopB–CTD that has the entire tetratricopeptide repeat (TPR) domain (amino acids 56–280) containing the putative peptide-binding pocket but lacks the N-terminal DNA-binding domain (amino acids 1–55). The RopB–CTD binds SIP in a sequence-specific fashion, albeit at lower affinity than the full-length RopB (Supplementary Fig. 5), and SIP binding does not induce RopB–CTD polymerization[37]. In the low-resolution crystal structure of apo RopB–CTD[40], ambiguity existed regarding the structural elements in the region in RopB containing amino acids 159–200. It could have been built either as single continuous α helix or as a helix–loop–helix with the first α helix containing amino acids 159–179 and the second α helix containing amino acids 181–200. Without exception, the analogous region was present as helix–loop–loop in the high-resolution structures of all RopB structural homologs[51–55]. Thus, owing to the difficulty in interpreting the low-resolution electron density map and the conformity of helix–loop–helix motif

arrangement with the high-resolution full-length structures of RopB structural homologs, the region containing amino acids 159–200 was modeled as helix–loop–helix in the apo RopB–CTD structure[40]. However, the electron density map obtained using the higher resolution data from RopB–CTD–SIP crystals indicated the presence of a continuous α helix containing amino acids 159–200 (Supplementary Fig. 6a). Thus, the region containing amino acids 159–200 of RopB was built as single α helix in the RopB–CTD–SIP structure (Supplementary Fig. 6b, c). A major difference between the two models is that the C-terminal half of one subunit containing amino acids 180–281 extends above the N-terminal half of the opposing subunit of a RopB–CTD dimer (Supplementary Fig. 6c). As a result, each super helix structure of a RopB–CTD dimer is formed by TPR motifs from both subunits compared to RopB structural homologs in which each super helix structure is formed by TPR motifs from the same subunit (Supplementary Fig. 6c)[51–55]. Nevertheless, the overall arrangement of structural elements and amino acids involved in RopB dimerization identified in the previous study remain unchanged in the RopB–CTD–SIP structure[40].

Each asymmetric unit has two subunits of RopB–CTD, and only one subunit in the asymmetric unit has defined electron density for SIP (Fig. 3a). Thus, one crystallographic dimer exists in peptide-bound form, whereas the second dimer is present in the apo form. We used the RopB–CTD dimer in peptide-bound state for further analyses. Comparisons of apo RopB–CTD (PDB code: 5DL2), SIP-free RopB–CTD in the RopB–CTD–SIP structure, and SIP-bound RopB–CTD structures indicated that SIP binding did not induce major structural changes in RopB–CTD (Supplementary Fig. 7a, b). The structure of RopB–CTD dimer in three different states can be superimposed to each other with a root mean square deviation (r.m.s.d.) of 0.5–0.7 Å.

SIP is oriented with its N-terminus facing the solvent-exposed exterior of the pocket, whereas the C-terminus is buried in the deep end of the pocket (Fig. 3a). The SIP-binding surface in RopB is formed by helix α6 of TPR 3, helix α8 of TPR 4, and the C-terminal capping helix α12. The binding surface is predominantly composed of hydrophobic and aromatic amino acids, and asparagines that are characteristic of TPR domains[42,56] (Supplementary Fig. 8a). The peptide-contacting face of helix α6 is lined by N152 and F155, and the surface of helix α8 facing SIP has R188, N192, I195, and Q199 (Fig. 3b). The side chains of M267, F268, Y271, and K278 of the capping helix α12 are positioned to interact with SIP (Fig. 3b). The SIP–RopB contacts can be classified into three categories: (i) the anchoring electrostatic interactions at the deep end of the pocket between R188 of helix α8 and the carboxyl oxygen moiety of the C-terminal L8 of SIP, (ii) stabilizing polar contacts between the side chains of R188 and N192 of helix α8, and the peptide backbone of SIP, and (iii) hydrophobic contacts between the side chains of SIP amino acids and the side chains of F155' (where ' indicates the amino acid from from the second subunit of RopB–CTD dimer) and V191 of helix α6, I195 of helix α8, and M267, F268, and Y271 of helix α12. However, no direct interaction was observed between the side chain of L6 of SIP (SIP-L6) and RopB. Given that RopB–CTD has decreased affinity for SIP compared to full-length RopB (Fig. 4a, Supplementary Fig. 5), and RopB–CTD–SIP was crystallized under nonoptimal pH (pH 7.5) for RopB–SIP interactions (Fig. 2d), it is likely that the structure of RopB–CTD–SIP represents a low-affinity state. Additional interactions promoting high-affinity SIP binding may occur in full-length RopB and/or under optimal pH conditions. Nevertheless, the structural observations in this study indicate that each amino acid of SIP except the side chain of L6 is required for sequence-specific SIP recognition by RopB.

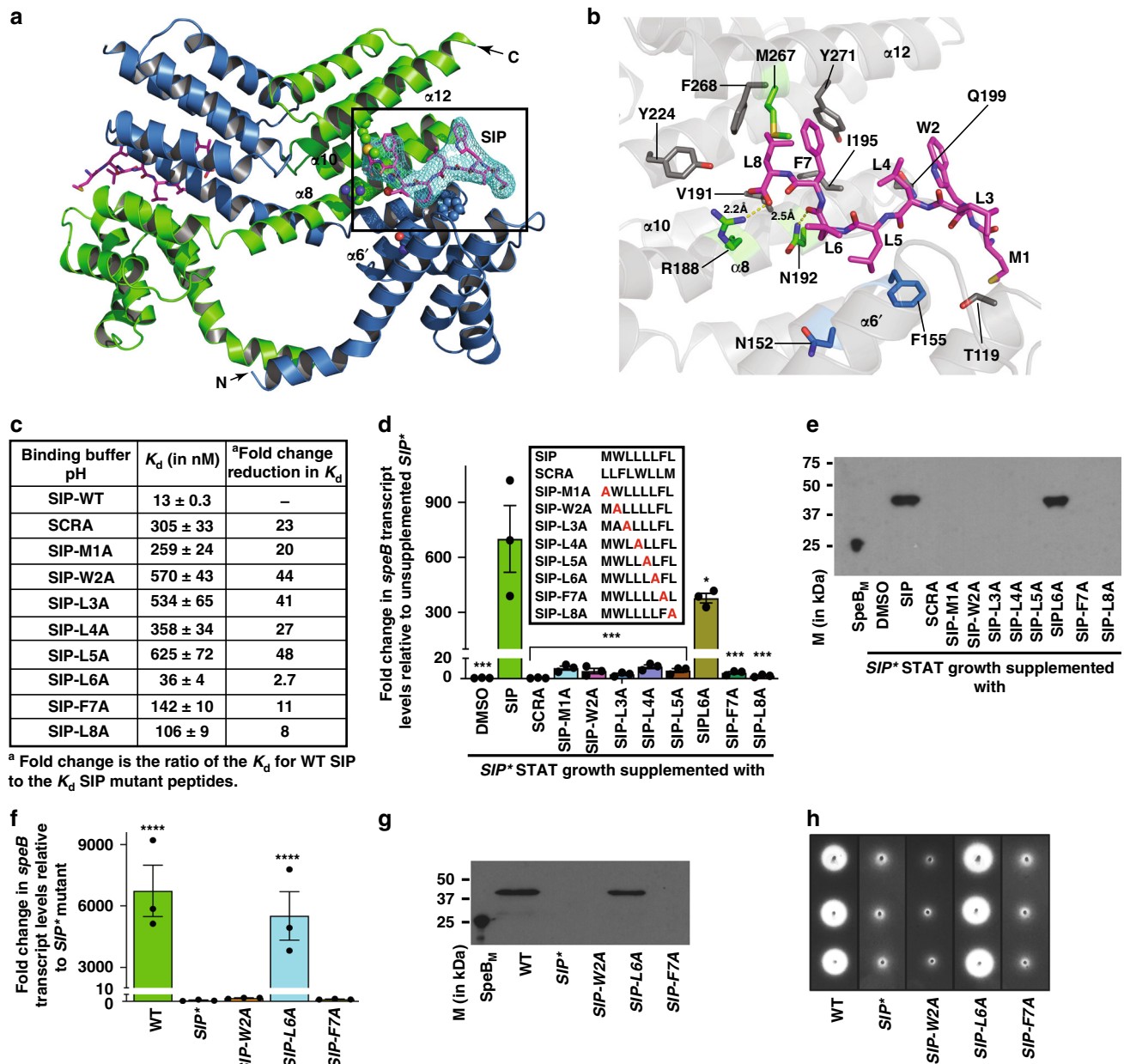

**Fig. 3** Molecular mechanism of SIP recognition by RopB. **a** Ribbon diagram of RopB–CTD dimer bound to SIP. Individual subunits of the dimer molecule are color-coded (blue and green). The 2Fo–Fc electron density map of SIP contoured at 1σ is shown. The SIP-binding pocket is boxed and labeled. The N- and C-termini of one subunit is marked as N and C, respectively. **b** Close up view of the interactions between RopB–CTD and SIP. SIP is shown as pink sticks and the eight amino acids of SIP are labeled. The SIP-interacting amino acid residues in RopB that are included in the mutational analyses from different subunits are colored in green and blue, respectively. The other SIP-contacting amino acid residues in RopB are colored in gray. The α-helices in RopB that form the SIP-binding pocket are labeled. **c** RopB–SIP-binding constants for synthetic SIP variants containing single alanine replacements at each position. **d** Analysis of the *speB* regulatory activity of synthetic SIP variants. The amino acid sequences of the synthetic peptides used in the experiment are shown in the inset. Scrambled peptide (SCRA) was used as a negative control. The *SIP** mutant strain supplemented with DMSO was used as a reference and fold-changes in *speB* transcript levels relative to the reference are shown. *P* values (***P < 0.001, *P < 0.05) of the indicated samples were compared to *SIP** mutant strain supplemented with WT SIP. **e** Western immunoblot analysis of secreted SpeB from indicated samples. Cell growth and synthetic peptide supplementation were performed as described in panel (**d**). Cell-free growth media (THY medium) were probed with anti-SpeB polyclonal rabbit antibody, and detected by chemiluminescence. The masses of molecular weight markers (M) in kilodaltons (kDa) are marked. Characterization of *SIP* mutant strains for *speB* gene transcript levels (**f**), secreted SpeB levels (**g**), and SpeB protease activity detected by casein plate assay (**h**). *P* values (****P < 0.0001) of the indicated strains were compared to *SIP** mutant strain. *P* values were determined by *t*-test. Source data for panels (**d**) and (**g**) are provided as a Source Data file

**Each SIP residue is critical for RopB binding and *speB* expression.** To probe the contribution of individual amino acids of SIP to sequence-specific recognition by RopB, we measured the binding affinities of synthetic SIP peptides containing single alanine substitutions at each position for RopB by FP assay (Fig. 3c). With the exception of SIP-L6A, the binding affinity of the mutant peptides for RopB was drastically and detrimentally affected (Fig. 3c and Supplementary Fig. 8b). SIP-L6A caused

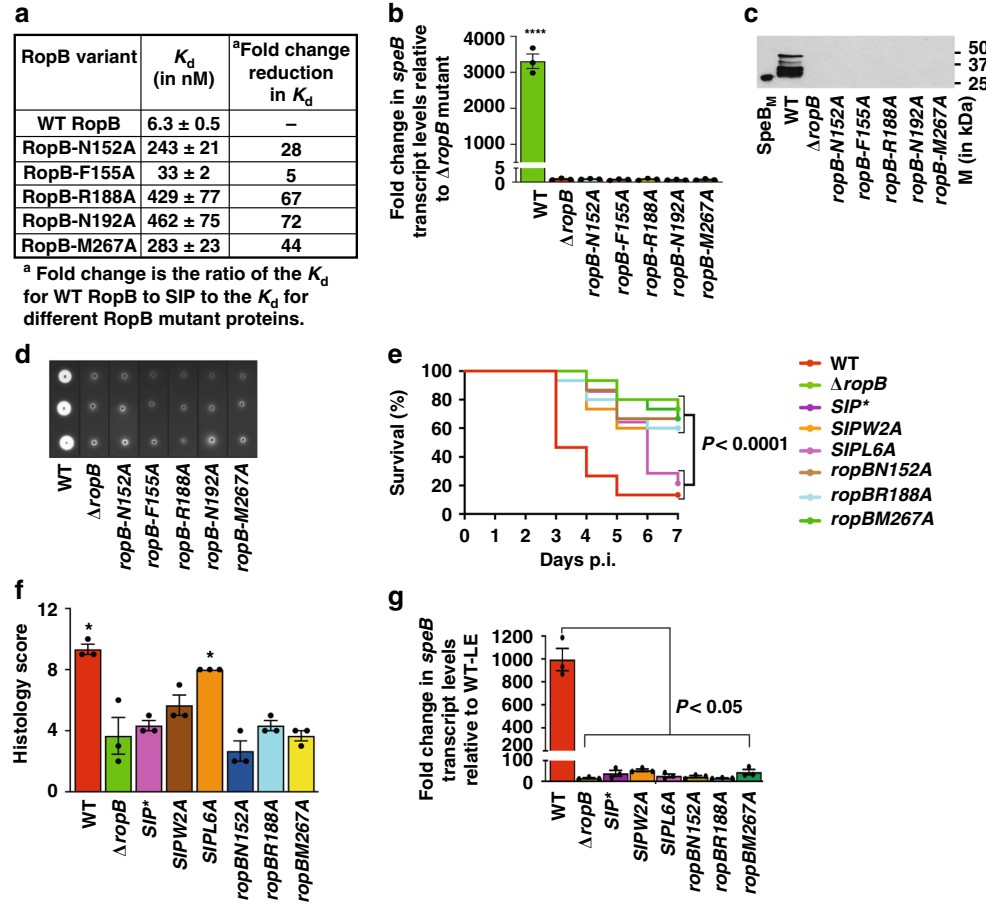

**Fig. 4** SIP-contacting residues in RopB are crucial for *speB* expression. **a** RopB–SIP-binding constants for recombinant RopB mutant proteins containing single alanine replacements in SIP-contacting amino acid residues. Characterization of isogenic *ropB* mutant strains for (**b**) *speB* gene transcript levels, (**c**) immunoreactive secreted SpeB levels, and (**d**) SpeB protease activity. *P* values (****$P < 0.0001$) of the indicated strains were compared to Δ*ropB* mutant strain. *P* values were determined by *t* test. Source data are provided as a Source Data file. **e** Twenty outbred CD-1 mice per strain were injected intramuscularly with $1 \times 10^7$ CFUs of each indicated strain. Kaplan–Meier survival curve with *P* values derived by log rank-test are shown. **f** Histopathology scores of hindlimb lesions from mice infected with each indicated strain. Histopathology analysis of the infected hindlimbs was performed at 48 h post-inoculation. Data are expressed as means + standard deviation. *P* values (*$P < 0.05$) of the indicated strains were compared to isogenic Δ*ropB* mutant strain. *P* values were determined by *t* test. **g** Analysis of the *speB* transcript level in the intramuscular lesions from mice infected with indicated strains. Samples were collected 24 h postinoculation from the lesions ($n = 5$ per strain) and analyzed in triplicate by qRT-PCR. The *speB* transcript levels in wild-type GAS grown in THY to late-exponential growth phase (WT-LE, $A_{600}$ ~1.5) was used as a reference and fold-changes in *speB* transcript levels relative to the reference are shown. *P* values were determined by *t* test

only relatively minor reduction (~2.7-fold) in the binding affinity for RopB compared to SIP (Fig. 3c and Supplementary Fig. 8b). SIP binding to RopB promotes high-affinity interactions between RopB and operator sequences in *speB* promoter[37]. Thus, we tested if defective SIP binding alters RopB–promoter interactions by FP assay using fluoresceinated oligoduplexes containing RopB binding sites from *speB* promoter[37]. We found that SIP variants that are defective in RopB binding disrupted RopB–promoter interactions (Supplementary Fig. 8c, d), whereas the SIP-L6A peptide caused WT-like binding to operator sequences (Supplementary Fig. 8c, d).

To correlate the binding affinities for RopB to SIP-dependent RopB regulatory activity, we performed a SIP addition assay using the *SIP** mutant strain. Consistent with the biochemical data (Fig. 3c), the SIP-L6A peptide had partial activity, whereas all other SIP mutant peptides lost their ability to activate RopB-dependent *speB* expression, and enhance the level of secreted SpeB (Fig. 3d, e). Furthermore, we generated isogenic *SIP* mutant strains containing single alanine substitutions at positions SIP-W2, SIP-L6, and SIP-F7 in the GAS genome. The isogenic

*SIP-L6A* mutant strain had WT-like phenotype, whereas the isogenic *SIP-W2A* and *SIP-F7A* mutant strains had drastically reduced *speB* expression, SpeB protease levels and enzymatic activity (Fig. 3f-h, Supplementary Fig. 8e). Collectively, these data are consistent with the interpretation that with the exception of L6, the side chain of each amino acid of SIP contributes to RopB–SIP interactions and RopB-dependent *speB* expression.

**SIP-contacting RopB residues are crucial for *speB* expression.** To determine the contribution of SIP-contacting residues in the TPR domain of RopB to SIP recognition, we introduced single alanine substitutions in RopB. We targeted the amino acids that are involved in anchoring (R188), hydrophobic (F155 and M267), and peptide backbone contacts (N192) with SIP for functional analysis. The side chain of N152 from the second subunit is not involved in direct contact with SIP in the RopB–CTD–SIP structure. However, N152 is highly conserved among RopB-like regulators (Supplementary Fig. 12), and the side chain of N152' is located on the peptide-facing surface of helix α6 from the second

subunit of RopB–CTD dimer. Thus, we hypothesized that N152 is involved in high-affinity RopB–SIP interactions under optimal binding conditions, and included N152 in the functional analysis. As demonstrated by their WT-like solubility, the single alanine substitutions did not affect recombinant RopB solubility or GAS viability (Supplementary Fig. 9a, c). FP assays using the purified recombinant RopB mutant proteins showed that the single alanine substitutions in SIP-contacting residues of RopB drastically reduced the binding affinity of SIP for RopB (Fig. 4a, Supplementary Fig. 9b).

Next, we assessed the role of SIP-contacting RopB amino acids in the regulatory activity of RopB. To test this, we generated isogenic mutant strains containing single alanine substitutions at the SIP-contacting amino acids in RopB. In accordance with the in vitro findings, the *speB* transcript level, secreted immunoreactive SpeB level, and SpeB protease activity were significantly reduced in the isogenic *ropB* mutant strains (Fig. 4b-d). Together, these results indicate that the SIP-specific recognition conferred by amino acids in the RopB TPR domain is critical for RopB-dependent *speB* expression.

**RopB–SIP interactions are critical for GAS virulence**. To test the hypothesis that gene regulation by RopB–SIP interactions is critical for GAS pathogenesis, we compared the virulence of the WT and isogenic *ropB* or *SIP* mutant strains in a mouse model of necrotizing myositis. Isogenic GAS mutant strains containing single alanine substitutions in RopB amino acids involved in anchoring contact (R188A), peptide backbone contact (N152A), and hydrophobic interactions (M267A) with SIP were used. Similarly, we assessed the virulence phenotype of an inactive (*SIP-W2A*) and an active *SIP* mutant (*SIP-L6A*) strain. With the exception of the *SIP-L6A* mutant, the isogenic *ropB* and *SIP* mutant strains were significantly less virulent than the WT parental strain and comparable to that of *ΔropB* and *SIP** mutant strains (Fig. 4e). As expected, the *SIP-L6A* mutant strain had WT-like virulence phenotype (Fig. 4e). Inasmuch as SpeB contributes to host tissue damage and disease dissemination, we compared lesion character by microscopic examination. Consistent with the virulence phenotype, relative to the WT strain, the *SIP-W2A* and *ropB* isogenic mutant strains caused smaller muscle lesions with less severe tissue destruction (Fig. 4f). The muscle lesions caused by the *SIP-L6A* mutant strain were equivalent to those caused by the WT (Fig. 4f). Finally, we investigated if the RopB–SIP interactions alter *speB* expression in vivo during the course of infection by measuring *speB* transcript levels in the infected lesions. Compared to the WT strain grown to LE phase in THY, the WT strain isolated from infected lesions had an 800-fold higher level of *speB* transcript (Fig. 4g). Consistent with the in vitro observations, lesions from mice infected with the isogenic *ropB* or *SIP* mutants had drastically decreased *speB* expression in vivo (Fig. 4g). Consistent with the delayed onset of mortality, the *SIP-L6A* mutant strain also had significantly decreased *speB* transcript levels at 24 h postinoculation (Fig. 4e, g). Together, these virulence data demonstrate that RopB–SIP interactions occur in vivo and single alanine substitutions affecting these interactions significantly attenuate GAS virulence.

**pH sensing and pH-dependent virulence regulation by RopB**. Our structure–function analyses of RopB–SIP interactions identified the amino acids involved in SIP recognition by RopB. However, the structural basis for pH-dependent SIP binding and gene regulation by RopB was not evident. High-affinity SIP–RopB interactions and SIP signaling occur at maximal levels between pH 5.5 and 6.0 (Fig. 2). Inasmuch as the histidine side chains have a *pKa* of 6.2, we hypothesized that pH-dependent protonation of

histidine(s) in a functionally important region of RopB influences high-affinity SIP binding and *speB* expression. Six (H81, H93, H144, H265, H266, and H277) of the seven total histidines in RopB are located in the CTD and one (H12) is in the DNA-binding domain (Fig. 5a, Supplementary Fig. 10a). With the exception of H144, all histidines in the CTD are surface-exposed and not located in known functional domains of RopB (Fig. 5a). Amino acid H144 is located at the base of the SIP-binding pocket and thus is ideally positioned to influence SIP binding indirectly (Fig. 5a). The side chain of H144 is engaged in intramolecular interaction with the side chain of Y176 (Fig. 5a, b). In the structure of RopB–CTD crystallized at pH 7.5, the side chains of Y182' and E185' (where ' indicates the amino acids from the second subunit of a RopB–CTD dimer) are oriented toward H144 but located farther (5.8 and 4.9 Å, respectively) to interact with H144 (Fig. 5b). However, protonation of H144 under acidic pH may bring them closer and facilitate the interactions between H144, Y182', and E185' suggesting that H144 may be involved in pH sensing.

To test the significance of histidines in RopB for gene regulation, isogenic mutant strains containing single alanine substitutions at histidines were generated and tested for SpeB protease activity. The isogenic H12A mutant strain lost SpeB protease activity (Fig. 6a). However, the H12A mutant protein had WT-like SIP binding but drastically reduced ability to bind *speB* promoter, suggesting that defective protease production by the isogenic H12A mutant strain is due to the role of H12 in DNA binding (Fig. 6b and Supplementary Fig. 10b–d). Among the histidines in the CTD, only the mutant strain with the H144A replacement was defective in SpeB protease production (Fig. 6a). To determine the functional role of H144-mediated intramolecular interaction network, we generated isogenic mutant strains with single alanine substitutions at contacting residues (Y182 and E185) and two noncontacting residues (Y181 and H277). These substitutions do not cause protein misfolding or affect GAS growth in vitro (Supplementary Fig. 10b, e). However, disruption of the intramolecular interactions by single alanine substitutions at contacting residues (H144A, Y182A, and E185A) impaired RopB–SIP interactions, *speB* expression, immunoreactive SpeB protein levels and SpeB protease activity (Fig. 6a–d). Conversely, isogenic mutant strains with single alanine substitutions at noncontacting residues had a WT-like phenotype (Fig. 6a–d).

These observations led us to hypothesize that pH-induced protonation of H144 at pH 6 and its interactions with Y176, Y182', and E185' increase the stability of RopB. To test this hypothesis, we compared the melting temperatures ($T_m$) of WT and H144A mutant proteins under different pH. Consistent with our hypothesis, WT RopB had a remarkable pH-dependent increase in stability. The initial unfolding temperature of RopB increased by 11 °C in pH 6 (31 °C) compared to above neutral pH (20 °C) (Fig. 6e–h). Similarly, the $T_m$ of WT RopB increased by 6.4 °C in pH 6 ($T_m$ 43.8 °C) compared to above neutral pH ($T_m$ 37.4 °C) (Fig. 6e, h, Supplementary Fig. 10f). However, no pH-dependent stabilization was observed for the H144A mutant protein (Fig. 6f–h, Supplementary Fig. 10g), suggesting that the intramolecular interactions formed between H144, Y182', and E185' in pH 6 contribute to the pH-dependent stabilization of RopB. To ensure that the observed pH-dependent stabilization of RopB is specific for H144, we measured the melting temperatures of mutant proteins containing single alanine substitutions at two non-pH sensing histidines, H12 from the DNA-binding domain and H277 from the C-terminal domain. Consistent with our hypothesis that H144 is the pH sensor in RopB, H12A, and H277A mutant proteins were pH sensitive and had a WT RopB-like pH-dependent increase in stability (Supplementary Fig. 11). Together, these structural and biochemical data demonstrate that

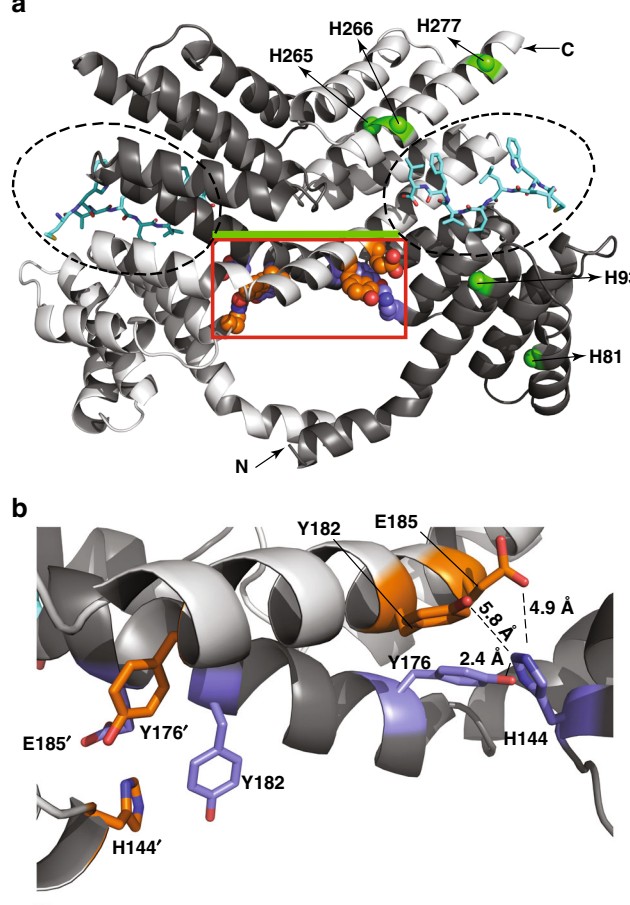

**Fig. 5** A histidine switch in RopB senses environmental pH. **a** Individual subunits of RopB–CTD dimer are color-coded in dark and light gray. The N- and C-termini of one subunit is marked as N and C, respectively. The two SIP-binding pockets in each subunit of a RopB–CTD dimer are circled (dotted lines). The green line connecting the two SIP-binding pockets indicates the location of the base of the SIP-binding pocket. SIP located in the peptide-binding pockets of the RopB–CTD dimer are shown as sticks and colored in cyan. The main chain atoms of surface-exposed histidines in one subunit of RopB–CTD are shown as green spheres and labeled. The side chains of H144, Y176, Y182', and E185' located at the base of the SIP-binding pocket for each subunit of a RopB–CTD dimer are shown as spheres and boxed in red rectangle (and in panel **b**). The ' indicates the amino acid residue from the second subunit of a RopB–CTD dimer. The side chains of the amino acid residues involved in intramolecular interactions from two subunits of a RopB–CTD dimer are color-coded in orange and purple, respectively. **b** A magnified view of the intramolecular interactions at the base of SIP-binding pocket of RopB in the boxed area (red) in panel **a**. The side chains of H144, Y176, Y182', and E185' located at the base of the SIP-binding pocket for each subunit of a RopB–CTD dimer are shown as sticks and the side chains from two subunits are color-coded in orange and purple, respectively. The amino acid residues from the second subunit of RopB–CTD dimer are indicated by '. The distances (in angstroms, Å) between the amino acid residues are shown

the H144-induced pH-sensing intramolecular interactions in RopB are critical for SIP-dependent *speB* expression.

**The pH-sensing RopB histidine switch is crucial for GAS virulence.** To investigate the significance of the pH-sensitive intramolecular interactions in RopB to GAS virulence, we assessed the virulence of isogenic H144A and Y182A mutant strains in a mouse model of necrotizing myositis. As anticipated, the

isogenic H144A and Y182A mutant strains were significantly less virulent than the WT and comparable to that of Δ*ropB* mutant strain (Fig. 6i–k). Furthermore, comparison of lesion character by visual and microscopic examination showed that isogenic *ropB* mutant strains, H144A and Y182A, caused smaller muscle lesions with less severe tissue destruction relative to WT (Fig. 6j, k). Collectively, these data demonstrate that the amino acids participating in the pH-sensing intramolecular interactions in RopB are critical for GAS virulence.

## Discussion

Here, we report that GAS uses a complex interplay between endogenous (SIP) and an environmental signal (acidification) additively to coordinate virulence factor production and influence pathogenesis. Our data show that GAS has integrated the pH-sensing mechanism into the SIP signaling pathway through a histidine switch in the cytosolic quorum-sensing regulator, RopB. A pH-sensitive histidine (H144) in RopB senses the environmental acidification and likely induces pH-dependent reorganization of intramolecular interactions at the base of SIP-binding pocket. Subsequently, the proposed pH-induced allostery in RopB promotes high-affinity SIP binding in the peptide-binding pocket of RopB, and triggers SIP-dependent upregulation of *speB* expression. The pH dependence of bacterial quorum sensing pathways also occurs in the competence regulatory pathways of *S. mutans* and pneumococci as well as in the *agr* virulence regulatory pathway in *Staphylococcus aureus*[11–15,57]. It is plausible that the histidine-dependent pH sensing, and the complex interplay between pH and quorum sensing pathways also occur in other microbial signaling pathways. Thus, our delineation of the mechanistic and regulatory details of the cross talk between pH sensing and quorum sensing may provide a basis for understanding environment sensing and gene regulation in other microorganisms, pathogenic, and otherwise.

Gene regulation by pH-sensing histidine switches has been identified in the sensor kinases of bacterial two-component signaling pathways[18,58,59]. However, unlike the extracellular kinases, the histidine switch identified here controls the regulatory activity of an intracellular quorum-sensing regulator. The environmental pH sensing and pH-dependent gene regulation by intracellular RopB requires that the GAS cytosol is acidified during bacterial growth. In this regard, we note that several lines of experimental evidence suggest that GAS cytosolic acidification occurs. GAS encounters an acidified extracellular environment in the infected tissue due to auto-acidification and/or abscess development[60–63]. Further, unlike nonlactic acid bacteria, the lactobacilli and streptococci lack elaborate pH homeostasis mechanisms[64,65]. As a result, the decrease in extracellular pH causes cytosolic acidification[64–66]. Typically, the lactic acid bacteria maintain a pH difference ($\Delta pH = pH_{intracellular} - pH_{extracellular}$) of 0.5–0.8 units. Consistent with this, GAS cytosol has a $\Delta pH$ of 0.2–0.3 units during environmental acidification (Fig. 2e, f). Thus, when the environmental pH decreases to 5.5, as observed in vitro during high GAS population density (Fig. 1a), intracellular acidification (pH ~5.8) that is optimal for histidine protonation occurs (Fig. 2e, f). Therefore, it is likely that environmental pH is a key physiological signal controlling the regulatory activity of RopB. Importantly, the amino acids involved in pH sensing and pH-dependent intramolecular interactions are conserved in RopB-like regulators in other gram-positive bacteria (Supplementary Fig. 12). We speculate that similar pH-sensing intracellular histidine switches are operative in other microbial signaling pathways and coupling the pH sensitivity of histidine to peptide binding and gene regulation is a general allosteric strategy employed by other bacteria.

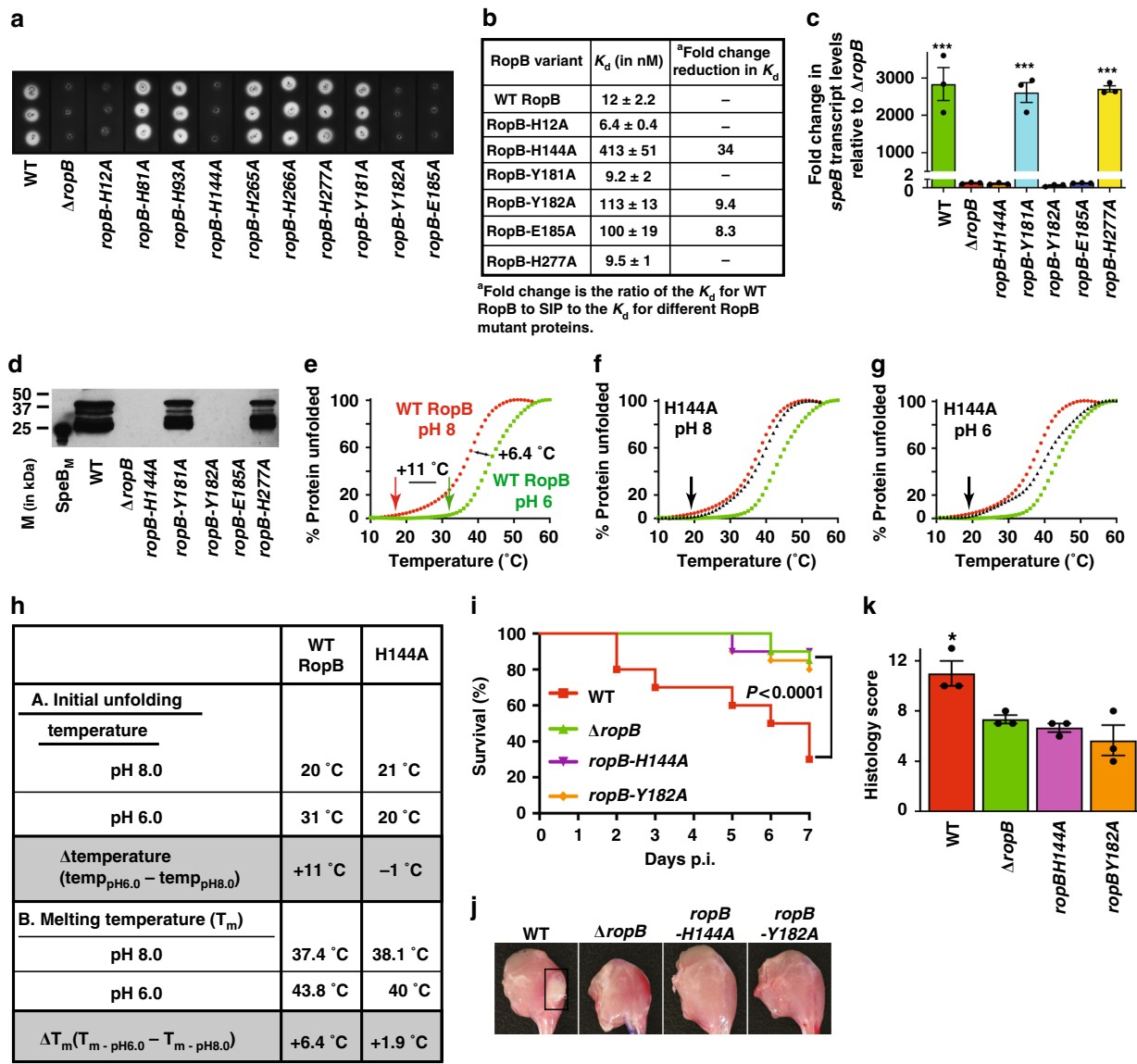

**Fig. 6** The histidine switch in RopB is critical for *speB* expression and GAS virulence. **a** SpeB protease activity made by each isogenic *ropB* mutant strain. **b** RopB–SIP-binding constants for the indicated recombinant RopB mutant proteins. Characterization of *ropB* mutant strains for *speB* transcript levels (**c**), and secreted SpeB levels (**d**). *P* values (***$P < 0.001$) of the indicated strains were compared to Δ*ropB* mutant strain. *P* values were determined by *t*-test. Source data are provided as a Source Data file. **e** Thermal stability of WT RopB was determined by a thermofluor assay. Thermal shift assay results for WT RopB in below (pH 6, colored green) and above neutral pH (pH 8, colored red) are shown. The temperature at which initial unfolding of WT RopB occurs are marked by vertical arrows and color coded. The horizontal two-headed arrows indicate the differences in melting temperatures ($T_m$) and in temperatures at which initial unfolding occurs between below and above neutral pH. The thermal stability curves of H144A mutant protein in above (pH 8) (**f**) and below (pH 6) (**g**) neutral pH are overlaid onto the thermal stability curves of WT RopB and colored in black. The temperature at which initial unfolding of H144A mutant protein occurs is marked by black vertical arrows. **h** Comparison of thermal shift assay results of WT RopB and H144A mutant protein. **i** Twenty outbred CD-1 mice per strain were injected intramuscularly with $1 \times 10^7$ CFUs of each indicated bacterial strain. Kaplan–Meier survival curve with *P* values derived by log rank-test are shown. **j** Analyses of gross hindlimb lesions from mice infected with each indicated strain. Analysis of the infected hindlimbs was performed at 48 h postinoculation. Larger lesion with extensive tissue damage in WT-infected mice in pH 6 is boxed (black box). **k** Histopathology scores of hindlimb lesions from mice infected with each indicated strain. Histopathology analysis of the infected hindlimbs was performed at 48 h postinoculation. Data are expressed as means + standard deviation. *P* values (*$P < 0.05$) relative to Δ*ropB* mutant strain are shown. *P* values were determined by *t* test

Our findings permit us to propose a model for *speB* regulation by the convergence of environmental pH and SIP (Fig. 7). At near-neutral environmental pH, the unprotonated H144 causes destabilization of its intramolecular interactions with Y176, Y182', and E185' located at the base of the SIP-binding pocket (Fig. 7, left). As a result, the environmental pH during low-GAS population density disfavors the intracellular high-affinity RopB–SIP interactions resulting in the loss of SIP autoinduction

and *speB* expression (Fig. 7). Conversely, during high-GAS population density, environmental acidification leads to lowering of GAS cytosolic pH. The intracellular acidic pH promotes protonation of H144 and stabilization of the networking interactions among H144, Y176, Y182', and E185' (Fig. 7, right). The pH-sensitive intramolecular interactions promote high-affinity RopB–SIP interactions. Thus, the regulatory influence of environmental pH occurs upstream of SIP as pH modulates SIP

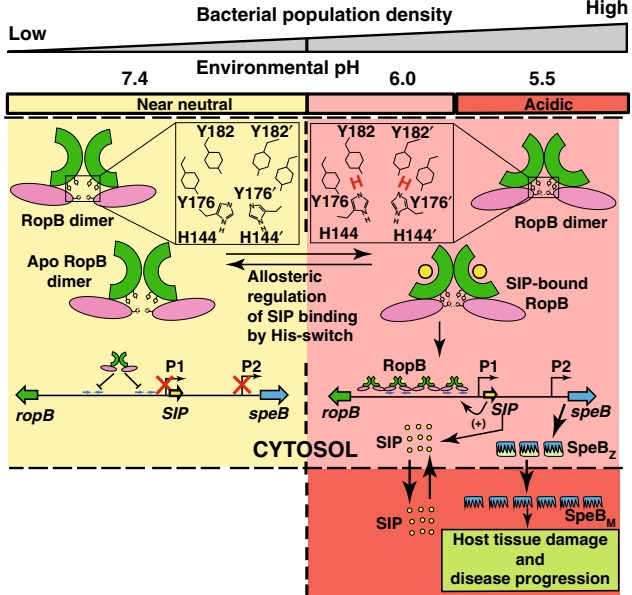

**Fig. 7** Model of GAS virulence regulation by environmental pH and SIP. At low-bacterial population density and near-neutral environmental pH (left panel), the deprotonated side chain of H144 destabilizes the intramolecular interactions with Y176, Y182′, and E185′. The weakened interactions at the base of the SIP-binding pocket inhibit high-affinity RopB–SIP interactions resulting in defective RopB–DNA interactions and decreased RopB-dependent transcription activation of *SIP* and *speB*. At high-population density (right panel), environmental pH decreases to pH 5.5, resulting in acidification of the GAS cytosol. When the intracellular pH becomes closer to the *pKa* of histidine (pH ~6.2), the protonated side chain of RopB H144 facilitates the interactions with Y176, Y182′, and E185′. The stabilized intramolecular interactions at acidic pH promote high-affinity RopB–SIP interactions. The high-affinity RopB–DNA interactions and RopB polymerization aided by SIP binding leads to upregulation of *SIP* expression, which then triggers robust induction of SIP production by a positive feedback mechanism. Finally, SIP-dependent upregulation of *speB* results in secretion of SpeB zymogen (SpeBZ). The acidified extracellular environment promotes rapid maturation of SpeBZ to SpeBM, and maximal protease activity of SpeBM, facilitating disease progression by cleaving various host and GAS proteins[24]

recognition by RopB. The productive association between RopB and SIP leads to the activation of the SIP autoinduction circuit, and subsequent upregulation of *SIP* and *speB* expression. As a result, environmental acidification coupled with increased SIP production converge to upregulate *speB* expression and increase virulence by influencing SIP signaling at two different steps: (i) SIP recognition by RopB and (ii) SIP production by controlling the positive feedback loop that couples SIP sensing by RopB to *SIP* expression. Finally, production of SpeB protease under acidic environmental pH promotes accelerated maturation of secreted zymogen into active protease resulting in increased proteolytic cleavage of substrates by mature SpeB (Fig. 7).

To summarize, the data we present here identify a two-pronged sensory mechanism in a quorum-sensing regulator of a human pathogen that allows the bacteria to perform effective sampling of the environment and orchestrate virulence factor production in an environment conducive for its maximal activity. The demonstration of convergence of two disparate signals, namely environmental pH and population density-specific chemical signals, in a bacterial quorum-sensing pathway to control virulence regulation not only provides novel insights into complexities of bacterial signaling but also suggest unique pH-based therapeutic possibilities to combat bacterial infections.

## Methods

**Bacterial strains, plasmids, and growth conditions.** Bacterial strains and plasmids used in this study are listed in Supplementary Table 1. Strain MGAS10870 is a previously described invasive serotype M3 isolate whose genome has been fully sequenced[67]. MGAS10870 is representative of serotype M3 strains that cause invasive infections and has wild-type sequences for all known major regulatory genes[67]. *Escherichia coli* DH5α strain was used as the host for plasmid constructions and BL21(DE3) strain was used for recombinant protein overexpression. GAS was grown routinely on Trypticase Soy agar containing 5% sheep blood (BSA; Becton Dickinson) or in Todd–Hewitt broth containing 0.2% (w/v) yeast extract (THY; DIFCO). When required, kanamycin or ampicillin was added to a final concentration of 50 or 100 µg/ml, respectively. Chloramphenicol was used at a final concentration of 15 µg/ml. All GAS growth experiments were done in triplicate on three separate occasions for a total of nine replicates. Overnight cultures were inoculated into fresh media to achieve an initial absorption at 600 nm ($A_{600}$) of 0.03. Bacterial growth was monitored by measuring the absorption at 600 nm ($A_{600}$). The *E. coli* strain used for protein overexpression was grown in Lysogeny broth (LB broth; Fisher).

**Correlation of GAS growth kinetics with *speB* expression and growth medium pH.** GAS was grown overnight in Todd–Hewitt broth supplemented with 0.2% yeast extract (THY; BD Biosciences, Sparks, MD), diluted 1:100 with fresh THY and grown to indicated growth phase. Aliquots were removed at the indicated time points. The pH of growth medium, and absorbance at wavelength 600 nm ($A_{600}$) were determined. The bacterial cells aliquoted at the indicated time points were incubated with RNAprotect, and cell pellets were processed to assess *speB* transcript levels by qRT-PCR.

**Measurement of intracellular pH.** The cytosolic pH ($pH_i$) was determined based on the previously described fluorescent probe method[48–50]. Cells grown to mid-exponential phase of growth ($A_{600}$ ~0.5) in THY were harvested by centrifugation, washed two times in 150 mM NaCl, and suspended in 50 mM HEPES buffer (pH 8.0). The cells were then incubated for 20 min at 37 °C in the presence of 10 µM carboxyfluorescein diacetate succinimidyl ester (cFDASE, Invitrogen). cFDASE is hydrolyzed to carboxyfluorescein succinimidyl ester (cFSE) in the cell and subsequently conjugated to aliphatic amines of the intracellular proteins. After incubation, cells were washed and suspended in 50 mM potassium phosphate buffer (pH 7.5). To eliminate nonconjugated cFSE, cells were incubated with 10 mM glucose for 30 min at 30 °C. Subsequently, cells were washed twice and suspended in 150 mM NaCl. Equal amount of cFSE-loaded cells were suspended in 0.5 ml of each buffer with indicated pH. After incubation in the indicated pH conditions for 5 min, fluorescence intensities were determined with an excitation spectrum of 400–500 nm wavelength range that includes excitation wavelengths 490 nm (pH-sensitive) and 435 nm (pH-insensitive). Emission was determined at 520 nm. The ratio of the emission resulting from excitation at 490 and 435 nm obtained for both cell suspension (*C*) and filtrate (*F*) was calculated as $R_{490/435} = (C_{490} − F_{490})/(C_{435} − F_{435})$. A calibration curve was determined in potassium phosphate buffers adjusted to pH values ranging from 5.5 to 8.0 and a cubic equation for the ratio value was determined. GAS intracellular pH values were calculated using the cubic equation from the calibration curve.

**SpeB overexpression and purification.** The coding region of *speB* of strain MGAS10870 without its secretion signal sequence (amino acids 1–27) was cloned into plasmid pET-28a. Site-directed mutagenesis was carried out to introduce serine substitution at C192 of SpeB. The primers used to generate C192S mutant are listed in Supplementary Table 2. Protein was overexpressed in *E. coli* strain BL21(DE3). Cells were grown at 37 °C till the $A_{600}$ reaches 0.5 and induced with 0.5 mM IPTG at 37 °C for 3 h. Cells were resuspended in buffer A (20 mM Tris HCl pH 8.0, 0.2 M NaCl, 5% glycerol, and 1 mM TCEP) and lysed by a cell lyser (Constant systems). The N-terminal hexa-histidine tagged zymogen form of SpeB was purified by affinity chromatography using a Ni-NTA agarose column. Purified recombinant SpeB zymogen was used to perform auto-activation experiments. To obtain the mature form of SpeB (SpeBM), SpeB zymogen was incubated at 4 °C for 2 days to facilitate its autocatalytic conversion into SpeBM. Finally, SpeBM was purified by size exclusion chromatography with superdex 75G column. The protein was purified to >95% homogeneity and the sequence identity of the purified SpeBM was confirmed by mass spectrometry-based identification of the N-terminal amino acids.

**SpeB auto-activation and protease activity assay.** Auto-activation of the WT SpeB zymogen into mature SpeB was performed by incubating 0.25 mg/ml of the zymogen at 37 °C in different buffer pH conditions. Aliquots were collected at indicated time points and reaction was stopped by adding SDS gel-loading buffer. Proteolytic processing of zymogen into mature form was visualized by sodium dodecyl sulfate polyacrylamide gel electrophoresis (SDS-PAGE).

To monitor the protease activity of the mature SpeB, the purified zymogen form of protease inactive SpeB mutant (C192S) was used as a substrate. Purified C192S was added to a final concentration of 0.25 mg/ml in reaction mixture containing 0.1 µg of mature SpeB in different buffer pH conditions. Aliquots were collected at indicated time points and reaction was stopped by adding SDS gel-loading buffer. Proteolytic processing of C192S zymogen into mature form was visualized by SDS-PAGE.

**RopB overexpression and purification**. The coding regions of the full-length and C-terminal domain (RopB–CTD) (amino acids 56–280) of *ropB* gene of strain MGAS10870 were cloned into plasmids pET-28a and pET-21b, respectively. Protein was overexpressed in *E. coli* strain BL21 (DE3). Protein overexpression and purification for both full-length RopB and RopB–CTD were carried out as described previously[37,68]. The proteins were purified to >95% homogeneity and concentrated to a final concentration of ~ 20 mg/ml.

**Crystallization and structure determination of RopB–CTD–SIP complex**. To prepare the SIP-bound RopB–CTD complex, the synthetic SIP was dissolved in the 100% DMSO and slowly added to RopB–CTD to obtain a final RopB:SIP ratio of 1:10. The final concentration of DMSO was ~5%. After overnight incubation at room temperature, the complex was centrifuged to eliminate undissolved peptide and the supernatant was concentrated to the 15 mg/ml using an Amicon concentrator (Millipore). Crystallization of RopB–CTD was performed using the vapor diffusion method with the crystallization solution containing 2.7 M potassium formate, 0.1 M Tris pH 8.5, 1% PEG2000, 0.15 M potassium chloride, and 5 mM EDTA. Preliminary crystals were further optimized for diffraction quality using a reservoir solution containing 2.7 M potassium formate, 0.1 M Tris pH 7.5, 1% PEG2000, 0.15 M potassium chloride and 1 mM EDTA. Sodium malonate to a final concentration of 90 mM was added to the drop as an additive. The diffraction data of the RopB–CTD–SIP crystals were collected at the Advance Light Source (ALS) beam line 8.3.1 (Berkeley, CA) at a single wavelength (1.117 Å) at 100 K temperature. Data were processed with iMOSFLM[69] and SCALA[70]. Apo RopB–CTD structure (PDB: 5DL2)[40] was used to obtain the initial phase by molecular replacement. Iterated rounds of model building were done using "COOT"[71] and refinement of the built model was performed using Refmac5 and Phenix[72]. The quality of the final model was verified using Molprobity[73,74]. The final model contains 227 residues form chain A, 226 residues from chain B, and 13 water molecules in the asymmetric unit with 98.9% residues in the favored regions of the Ramachandran plot and 1.1% residues in the disallowed region. Selected X-ray data collection, phasing, and refinement statistics are given in Supplementary Table 3. All structure-related figures were generated using Pymol[75].

**Synthetic peptide addition assay**. Synthetic peptides of high purity (>90% purity) obtained from Peptide 2.0 (Chantilly, VA) were suspended in 100% DMSO to prepare a 10 mM stock solution. Stock solutions were aliquoted and stored at −20 °C until use. Working stocks were prepared by diluting the stock solution in 25% DMSO.

**Creation of isoallelic strains**. Isoallelic strains containing either single codon changes or inactivation of entire coding region were generated as previously described[76]. A DNA fragment with approximately 600 bp on either side of the coding region of interest was amplified using the primers listed in Supplementary Table 2 and cloned into the multi-cloning site of the temperature-sensitive plasmid pJL1055[77]. The resultant plasmids were electroporated into group A *streptococci* and colonies with plasmid incorporated into the GAS chromosome were selected for subsequent plasmid curing. DNA sequencing was then performed to ensure that no spurious mutations were introduced.

**Transcript level analysis**. GAS strains were grown to the indicated $A_{600}$ and incubated with two volumes of RNAprotect (Qiagen) for 10 min at room temperature. RNA isolation and purification were performed with an RNeasy kit (Qiagen). After quality control analysis, cDNA was synthesized from the purified RNA using Superscript III (Invitrogen) and Taqman qRT-PCR was performed with an ABI 7500 Fast System (Applied Biosystems). Comparison of transcript levels was performed by the $\Delta C_T$ method of analysis using *tufA* as the endogenous control gene[5,78]. The primers and probes used for qRT-PCR are listed in Supplementary Table 2.

**Western immunoblot analysis of SpeB in the culture supernatant**. Cells were grown to indicated growth phase and harvested by centrifugation. The cell-free culture supernatant was prepared by filtration with 0.22 μM membrane and the filtrate was concentrated twofold by speed-vac drying. Equal volumes of the samples were resolved on a 15% SDS-PAGE gel, transferred to a nitrocellulose membrane, and probed with polyclonal anti-SpeB rabbit antibodies. SpeB was detected with a secondary antibody conjugated with horseradish peroxidase and visualized by chemiluminescence using the SuperSignal West Pico Rabbit IgG detection kit (Thermo Scientific).

**SpeB protease activity assay**. Analysis of SpeB protease activity was assessed by casein hydrolysis and zone of clearance on skim milk agar plates. GAS growth was stabbed on milk agar plates and protease activity was analyzed following overnight incubation at 37 °C.

**Site-directed mutagenesis of RopB**. The quick change site-directed mutagenesis kit (Agilent) was used to introduce single amino acid substitutions within the *ropB* coding region in plasmid pET28a-*ropB* and mutations were verified by DNA sequencing. The primers used to introduce the substitutions are listed in Supplementary Table 2.

**Fluorescence polarization assay**. Fluorescence polarization-based RopB-ligand binding experiments were performed with a Biotek microplate reader (Biotek) using the intrinsic fluorescence of fluorescein labeled DNA or synthetic peptides. The polarization ($P$) of the labeled DNA or synthetic peptides increases as a function of protein binding, and equilibrium dissociation constants were determined from plots of millipolarization ($P \times 10^{-3}$) against protein concentration. For RopB–DNA-binding studies, 1 nM 5′-fluoresceinated oligoduplex in binding buffer (20 mM Tris–HCl pH 8.5, 200 mM NaCl, 1 mM TCEP and 25% DMSO) was titrated against increasing concentrations of purified RopB and the resulting change in polarization measured. Samples were excited at 490 nm and emission measured at 530 nm. The RopB-peptide-binding studies were performed in a peptide-binding buffer composed of 20 mM potassium phosphate pH 6.0, 75 mM Nacl, 2% DMSO, 1 mM EDTA and 0.0005% Tween 20. All data were plotted using KaleidaGraph and the resulting plots were fitted with the equation $P = \{(P_{bound} - P_{free})[protein]/(K_D + [protein])\} + P_{free}$, where $P$ is the polarization measured at a given protein concentration, $P_{free}$ is the initial polarization of the free ligand, $P_{bound}$ is the maximum polarization of specifically bound ligand and [protein] is the protein concentration. Nonlinear least squares analysis was used to determine $P_{bound}$, and $K_d$. The binding constant reported is the average value from at least three independent experimental measurements.

**Fluorescence measurements**. To demonstrate the cytosolic internalization of exogenously added FITC-SIP, GAS cells were grown to early stationary phase of growth ($A_{600}$—1.7; $3.04 \times 10^8$ colony-forming units (CFUs)/ml) and incubated with the indicated concentrations of FITC-SIP for 1 h at 37 °C. Cells were harvested by centrifugation, washed twice with PBS, and resuspended in equal volume of PBS. Cells were lysed by fastprep (MP Biomedicals) and lysates were clarified by centrifugation at 13,000 rpm at 4 °C for 30 min. Samples were analyzed in 100 μl volume using an excitation and emission wavelengths of 490 and 520 nm, respectively. Readings were taken using a Biotek microplate reader (Biotek) and fluorescence measurements in relative fluorescence units were reported.

**Confocal fluorescence microscopy**. To demonstrate the internalization of FITC-SIP, GAS cells were grown to early stationary phase of growth ($A_{600}$—1.7; $3.04 \times 10^8$ CFUs/ml), suspended in each buffer with indicated pH, and incubated with FITC-SIP for 1 h at 37 °C. Cells were harvested by centrifugation, and washed twice with PBS. Cells were fixed on the coverslip using 1% glutaraldehyde and 3% formaldehyde. The images were taken using a Nikon Eclipse TiN-STORM super resolution microscope equipped with iXon3 897 EM-CCD camera.

**Animal virulence studies**. Mouse experiments were performed according to protocols approved by the Houston Methodist Hospital Research Institute Institutional Animal Care and Use Committee. These studies were carried out in strict accordance with the recommendations in the Guide for the Care and Use of Laboratory Animals, eighth edition. Virulence of the isogenic mutant GAS strains was assessed using intramuscular mouse model of infection (approved number AUP-0615-0041). For intramuscular infection, 10 female 3–4-week-old CD1 mice (Harlan Laboratories) were inoculated in the right hindlimb with $1 \times 10^7$ CFU of each strain and monitored for near mortality. Results were graphically displayed as a Kaplan–Meier survival curve and analyzed using the log-rank test. For histopathology, infected hindlimbs were examined at 48 h postinoculation. Tissues from excised lesions were fixed in 10% phosphate-buffered formalin, decalcified, serially sectioned, and embedded in paraffin using standard automated instruments. Hematoxylin and eosin and Gram's-stained sections were examined in a blinded fashion with a BX5 microscope and photographed with a DP70 camera (Olympus). Micrographs of tissues taken from the inoculation sites that showed pathology characteristic of each strain were selected for publication. Histology was scored by a pathologist blinded to the strain treatment groups as described previously[79]. Data are shown as means ± standard errors of the means (SEM), with statistical differences between strain groups determined using the Wilcoxon rank sum test.

**Quantitative bacterial culture from infected mouse tissue**. The infected limbs from 20 mice per group collected at 24 h postinoculation were used for quantitative culture. All tissues were weighed and homogenized with an OMNI homogenizer (USA Scientific Inc.). Tissue homogenates were diluted serially in sterile PBS and plated on TSA-B. The plates were incubated for 24 h at 37 °C in an atmosphere of 5% $CO_2$ and CFUs per gram of tissue were calculated. The results of each treatment group at each time point were expressed as mean ± SEM, and statistical significance was calculated by the Wilcoxon matched pairs test (Prism 4.03; GraphPad Software Inc.).

**Transcript analysis from infected tissue**. To assay in vivo transcript levels, skin or muscle lesions from four mice per infecting strain were collected 24 h post-infection and the tissue samples were incubated with RNAlater (Qiagen). Samples were snap frozen with liquid nitrogen and stored at −80 °C until use. RNA was isolated and purified using an RNeasy fibrous tissue mini kit (Qiagen). The quality and concentration of RNA were assessed with an Agilent 2100 Bioanalyzer. cDNAs were prepared using Superscript III (Invitrogen) and transcript levels were measured by Taqman qRT-PCR. Data were analyzed using the $\Delta C_T$ method.

**Thermofluor assay**. The thermal stability of purified WT RopB and H144A mutant proteins was compared using a differential scanning fluorescence (thermofluor) assay[80]. Purified recombinant WT or H144A proteins were added to achieve a final concentration of 0.1 mg/ml in buffers adjusted to indicated pH containing a 1:1000 dilution of Sypro orange (Invitrogen). A total of 40 µl of reaction mixture was added to a final volume of 300 µl in each well of a 96-well MicroAmp plate (ABI). The temperature was raised from 10 to 95 °C using a Bio Rad C1000 thermocycler. The data were fitted with GraphPad Prism version 7.0[81].

**Statistical analysis**. Prism (GraphPad Software 7.0) was used for statistical analyses. All GAS growth experiments for transcript level analyses were done in triplicate on three separate occasions for a total of nine replicates. The protein–peptide, protein–DNA binding and thermal stability experiments were done on three separate occasions to ensure reproducibility. For near mortality, values are shown as Kaplan–Meier survival curves, and statistical significance was determined using the log-rank test.

**Reporting summary**. Further information on research design is available in the Nature Research Reporting Summary linked to this article.

## Data availability

The coordinates and structure factors for the SIP-bound RopB–CTD structure have been deposited in the Protein Data Bank (PDB) with accession code 6DQL. Other data supporting the findings of this study are available in this article and its Supplementary Information files, or from the corresponding author upon request.

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

## Acknowledgements

This work was supported by National Institutes of Health grant 1R01AI109096-01A1 to M.K. and funds from the Fondren Foundation to J.M.M. H.D. was supported by the Basic Science Research Program through the National Research Foundation of Korea (NRF) funded by the Ministry of Education (2017R1A6A3A03008353). Advanced Light Source was supported by Department of Energy contract DE-AC03-76SF00098.

## Author contributions

H.D., N.M., A.R.V., M.O.S., R.J.O., and M.K. designed and performed research; H.D., N.M., R.J.O., and M.K. analyzed the data; H.D., J.M.M., and M.K. wrote the paper.

## Additional information

**Competing interests:** The authors declare no competing interests.

