## [Peer Review File · Nature Communications]

Reviewers' comments:

Reviewer #1 (Remarks to the Author):

Do et al. have made a significant and detailed experimentation of the confluence of pathways (pH and quorum sensing) that control expression of SpeB. The work make a significant and elegant contribution to the field. This work should also open up the way for other research groups investigating Gram +ve quorum sensing mechanisms.

The following points are provided for the authors consideration:

1. Did the authors bioinformatically examine other quorum sensing regulators (particularly if structure is known) for the presence of similar key motifs (histidine switch). This might be an informative aspect to include in the discussion of this work.
2. For Figure 1E, were the WT GAS present at significantly higher CFU loads compared to speB mutant GAS? At either/both pH?
3. Figure 2F. I had some initial difficulty interpreting the differences between extracellular and intracellular pH. It might be clearer for the reader if you use only 1 decimal point for the intracellular pH (same as extracellular) or you plot out as a bar graph presentation? What do you think is going on at pH8, where there is also a 0.3 pH change? Buffering capacity of the cell kicks in?
4. Figure 4G. The speB transcript levels were measured at 24 h and SIP L6A was no different to other mutants, and very distinct from WT. Do the authors think that if this experiment were conducted at a later timepoint (48 h or 72 h) that there would be a clear difference?
5. I would expect that the authors would have considered this point, but was there any attempt made to undertake the crystallisation study (RopB-CTD + SIP) at lower pH to directly visualize structural changes?
6. Bottom of page 15. The authors mention that the Y182 mutation did not result in protein misfolding. How was this accomplished? Was CD spectrometry (or other technique) undertaken to confirm?
7. For virulence experiments (Figure 4E and 6I), can the authors provide information in the Figure legends on the individual dose for each WT and mutant represented in the figure panels?

Finally, the authors work is exciting and novel. This paper provides a detailed model for the regulatory control of an important GAS virulence determinant.

Mark Walker

Reviewer #2 (Remarks to the Author):

In this manuscript, Do and colleagues explore how pH influences SpeB production via a quorum sensing (QS) peptide (SIP) interaction with the global regulator RopB. Notably, pH has been shown to control other QS pathways in other bacteria but this is the first study to discover specific molecular mechanisms that help explain this phenomenon. This study is detailed and happily, many times I asked an experimental question the authors went on to answer it. The real power in

the data is that it includes genetic, phenotypic, biochemical and in vivo demonstrations of pH regulation of SpeB production (via the SIP-RopB pathway). As mentioned before, there has been no obvious mechanism for pH-dependent control of QS in bacteria and this paper should spur others in the field to examine pH control of QS in other bacteria. My comments on the paper can be found below:

In some cases, statistics (as shown with *, etc) are used, and in other cases they are not. Examples where statistics may help follow: Figure 1B, Figure 2A, Figure 3D and 3F. Consistency would help the reader determine what is significant and what is not.

Figure S4B: The affinity for SIP binding to RopB is obviously increased at lower pH (reduced K_d). The dynamic range (total change in mP) is reduced with pH. Can the authors explain why this has happened?

Minor comments

The line "Each component of the SIP signaling pathway must be functional for a wild-type virulence phenotype" is repeated on page 4.

I appreciate that the authors have explored in great detail pH mediated regulation of SIP-RopB and have an excellent working model (Fig. 7). SpeB production is very complex and there is a limit to how much detail can be provided (a few excellent reviews are cited). However, do the authors believe that pH could influence other areas of the pathway e.g. the endopeptidase PepO, or CovRS?

Reviewer #3 (Remarks to the Author):

This review covers mainly the structural biology portion of the study. The authors previously determined the crystal structure of apo RopB. Here, they report the structure of RopB bound to its activating peptide SIP. The proposed model of RopB activation is moderately supported by the structural data:

- The proposed acidity-induced conformational change mediated by the histidine switch is not actually observed in this structure.
- The activating peptide seems to be bound to a low-binding-affinity conformation of the protein.
- The molecular mechanism by which SIP binding promotes RopB binding to DNA is still unknown, as the DNA-binding domain of the protein is absent from the structure.

On the other hand, the proposed model is more strongly supported by the accompanying biochemical and biological data.

Several issues need to be addressed:

1 - There seems to be a major error in the built structure of the protein (as well as in the authors' previously published structure of apo RopB in the same crystal form, PDB 5DL2). Two chains of RopB are found in the asymmetric unit (ASU), A and B. They are almost completely built but both are missing residue 180. The segments 159-179 and 181-200 form antiparallel helices, and in the current model, residue 180 appears to form a loop between the two helices. This helix-loop-helix segment is located very close to its crystallographic symmetry mate from the adjacent ASU (which also contains two chains of RopB, A' and B'). There is no electron density at the implied position of residue 180. However, there is clear continuous electron density from the helix 159-179 leading into helix 181'-200' of the adjacent ASU, and vice versa from helix 159'-179' of the adjacent ASU leading into helix 181-200. Thus, residues 159-200 form one long continuous helix, and half of the protein chain was built into the wrong ASU. Attached are pictures of the original and corrected arrangements. One full RopB chain is in green and its crystallographic

symmetry mate, in blue. Residues 179 and 181 are indicated by small and large spheres, respectively. This analysis applies to chain B as well. The model should be rearranged to correct this, or an explanation should be provided. This potential error actually has little impact on the structural analysis, but it would imply that the SIP binding site and the histidine switch are composed of residues from both copies of the crystallographic dimer. If the authors do apply this correction, they should also consider correcting their previously reported apo RopB structure.

2 - There are a few Ramachandran outliers or borderline values in the backbone angles of the SIP structure. Notably, a cis peptide bond was modeled at position 6-7. As the electron density for the peptide is not clearly defined at this resolution, this should very likely be corrected to a trans bond.

3 - Figure 3A:

Do the authors mean "composite omit electron density map"? Could they verify that this is indeed a 2fo- σ map displayed at the 1-sigma level? The map file that they provided, although being a refinement output map and not a composite omit one, shows weak fragmented density for the peptide at the 1-sigma level, and only shows a similar density to Figure 3A when contoured at 0.5-sigma.

4 - Section Structural Basis of SIP Recognition by RopB:

- "stabilizing polar contacts between the side chains of N152 of helix α 6, N192 of helix α 8, and K278 of helix α 12 of RopB and the peptide backbone of SIP"

The side chains of N152 and K278 are quite far from the peptide backbone, as seen in Figure 3B and in the coordinates file provided. Rather, the hydrogen bonds between RopB and SIP involve the side-chains of Y224 and N192, as listed correctly in Figure S6A. All the residues mentioned throughout this paragraph should be double-checked and Figure 3B modified accordingly.

- "hydrophobic contacts between the side chains of SIP amino acids and the side chains of F155 of helix α 6, I195 of helix α 8, and M267, F268, and Y271 of helix α 12"

V191 also establishes hydrophobic contacts with the peptide.

5 - Figures 4 and S7, and the related sections in the Results:

One of the mutants analyzed is N152A. However as mentioned in #4, N152 is quite far from the peptide (over 5 Angstroms from the L5 side chain and even further from the SIP backbone, as seen in Figure 3B and in the coordinates file provided). The authors should explain this or remove this mutant from the Results.

6 - Section Structural Basis of SIP Recognition by RopB:

"However, no direct interaction was observed between the side chain of L6 of SIP (SIP-L6) and RopB. Together, these data indicate that each amino acid of SIP except the side chain of L6 is required for sequence-specific SIP recognition by RopB."

Residues 1-4 of SIP establish few contacts with RopB, and those are weak non-specific Van der Waals interactions between hydrophobic side chains of the peptide and polar or charged residues of the protein. The structure alone does not seem to explain the importance of this portion of SIP for its recognition by RopB, although the biochemical and biological data do support the importance of all these residues. The binding affinity of SIP to RopB is around 10nM, and although the peptide was present at relatively high concentration during crystallization, its average crystallographic B-factor is two times higher than the protein's, indicating conformation heterogeneity or partial occupancy. Also, SIP is absent from the second RopB copy in the asymmetric unit. The authors propose a pH-induced conformational change in the protein that increases its affinity for the peptide. Based on all these points, have they considered the possibility that the current structure (at neutral pH) may represent a low-binding-affinity conformation in which RopB can still associate with SIP but without forming many of the specific interactions implied by their biochemical data that would result in nanomolar affinity? The authors should discuss this point if they deem it relevant.

7 - Related to #6, there should be a figure, at least in the supplementary, comparing the overall structure (for example in ribbon form) of the SIP-bound chain A crystallographic dimer with that of chain B and with the dimers from the previously published apo RopB structure. This would illustrate that the conformations of apo RopB and SIP-bound RopB from this neutral-pH structure are almost identical.

8 - Have the authors considered a potential role for residue E185 in the histidine switch? It is located close to H144, is highly conserved and could form a salt bridge with the latter at moderately acidic pH. The pH dependency of the interaction between H144 with Y182 is unclear, as tyrosine is uncharged and can act both as hydrogen bond donor and acceptor, and can participate in aromatic stacking. The data does implicate Y182, but there may be a larger conformational change at acidic pH, than only rotation of Y182 towards H144 or vice versa. In addition, if the authors implement the correction suggested in #1, this would mean that H144 interacts with a residue (Y182 or maybe E185) from the second subunit of the dimer, with possible implications in dimer formation or stability for the histidine switch.

9 - As the main novelty of the structure is the complex between RopB and SIP, it could be interesting to compare it to related proteins. For example, the TPR-containing, quorum-sensing regulator PlcR also undergoes a conformational change upon binding to a signaling peptide (PMID 23277548), albeit not pH-induced. Such a comparison could give clues as to how SIP binding to RopB stimulates DNA binding by the protein.

10 - Minor points:

- In the third paragraph of the introduction, this sentence is duplicated: "Each component of the SIP signaling pathway must be functional for a wild-type virulence phenotype"
- The Materials and Methods section is not very informative and could be re-written. Instead of listing every type of experiment followed by "details are provided in the supplementary", the authors should mention the key points here, and have one introductory sentence referring to the supplementary for the detailed Materials and Methods.
- Figure 3B: the gray side chains should have their oxygen and nitrogen atoms colored in red and blue, like the other residues in this panel.
- Figure 5A: the line connecting the base of the SIP-binding pocket of the two subunits does not stand out against the background of the rest of the image. The base of the pocket could be marked in some other way. Also, the residues forming the histidine switch, displayed as all-atom spheres, are difficult to distinguish from each other. They could be shown as sticks, or maybe one sphere per residue.
- Figure S8A: the source of the full-length RopB model is not indicated. Is it from a previous publication? If not, details should be provided in the supplementary methods.
- Table S3: the Wilson B-factor for the dataset should be provided in the Data Collection section.
- At several places in the manuscript, the authors refer to the pH-induced conformational change in RopB. As this rearrangement was not actually observed in the present structure, it should be specified as putative / proposed / probable / likely...

Alexei Gorelik, McGill University

Original

Corrected

Response to Reviewer's Comments:

Reviewer #1 (Remarks to the Author):

Do et al. have made a significant and detailed experimentation of the confluence of pathways (pH and quorum sensing) that control expression of SpeB. The work makes a significant and elegant contribution to the field. This work should also open up the way for other research groups investigating Gram +ve quorum sensing mechanisms.

Response: We thank the reviewer for judging our work as a significant and elegant contribution to the field. As noted by the reviewer, we also anticipate that our findings on the interplay between environmental pH and quorum sensing pathway is likely to initiate similar investigations in other bacterial quorum sensing mechanisms.

The following points are provided for the authors' consideration:

1. Did the authors bioinformatically examine other quorum sensing regulators (particularly if structure is known) for the presence of similar key motifs (histidine switch). This might be an informative aspect to include in the discussion of this work.

Response: RopB belongs to the RRNPP family of quorum sensing intracellular transcription regulators. Within the RRNPP family regulators, we have identified a subset of Rgg regulators that has the histidine switch conserved. We have discussed the findings from our bioinformatic analyses in the discussion (page 18, bottom of the page) and included the amino acid sequence alignment in the supplementary data (supplementary figure S11).

However, there are no published reports demonstrating the pH dependence of the quorum sensing pathways controlled by the structurally characterized RRNPP family regulators. Furthermore, our analyses of Rgg, PlcR, and PrgX (or NrpR) structures did not identify histidines (similar to H144 in RopB), tyrosines (similar to Y182 in RopB) or

glutamates (similar to E185 in RopB) at the location identified in RopB structure. However, if these pathways are shown to be pH dependent, RopB-like histidine-based pH sensing mechanism is still possible. But the pH-sensing histidine switch may reside in a location in the structure that is different from that of RopB, and controls peptide sensing or gene regulation by allosteric mechanisms that are distinct from RopB. Thus, our elucidation of pH-sensing histidine switch is likely to provide the foundation for such future studies.

2. For Figure 1E, were the WT GAS present at significantly higher CFU loads compared to *speB* mutant GAS? At either/both pH?

Response: We have observed significantly higher bacterial burden (higher CFUs) for WT GAS in both pH compared to the Δ *speB* mutant strain. However, we did not observe any pH-specific differences in the *in vivo* survival of individual strains (Fig. 1e). Importantly, compared to WT, the alterations in the inoculum pH did not affect the virulence phenotype of Δ *speB* mutant (Fig. 1c-d), demonstrating a direct correlation between increased SpeB production in SpeB-producing pH to early onset of lesion development.

3. Figure 2F. I had some initial difficulty interpreting the differences between extracellular and intracellular pH. It might be clearer for the reader if you use only 1 decimal point for the intracellular pH (same as extracellular) or you plot out as a bar graph presentation? What do you think is going on at pH8, where there is also a 0.3 pH change? Buffering capacity of the cell kicks in?

Response: We apologize for the inconvenience. In accordance with the reviewers first comment, we have revised the figure to include only one decimal point for the intracellular pH for better comparison with extracellular pH (Fig. 2f). We have also placed

the extracellular and intracellular pH measurements adjacent to each other in the table to facilitate the data analyses.

As the reviewer noted in the second comment, we also believe that GAS pH homeostasis mechanisms are activated in above neutral extracellular pH. As a result, GAS is able to maintain the cytosolic pH close to neutral pH in response to above neutral extracellular pH (pH 8.0).

4. Figure 4G. The *speB* transcript levels were measured at 24 h and SIP L6A was no different to other mutants, and very distinct from WT. Do the authors think that if this experiment were conducted at a later timepoint (48 h or 72 h) that there would be a clear difference?

Response: Our *in vitro* data on SIP-L6A offer clues to explain the *in vivo* phenotype. As shown in Figs. 3c and d, compared to WT SIP, synthetic SIP-L6A had reduced affinity for RopB (Fig. 3c) and SIP-L6A supplementation caused only partial induction of *speB* expression *in vitro* (Fig. 3d). These results suggest that SIP-L6A is less efficient in driving SIP-mediated *speB* expression than WT SIP. Such delicate differences, in part, explain the decreased *speB* transcript levels *in vivo* (Fig. 4g), and delayed mortality in mouse infection studies with *SIP-L6A trans-complemented strain* (Fig. 4e). However, as the reviewer indicated, we anticipate that if infected tissues collected at 72 h post-inoculation or later were analyzed, the SIP-L6A is likely to have higher levels of *speB* expression than the levels observed at 24 h post-inoculation or in defective *SIP* mutants. Given that the *in vitro* and *in vivo* data on SIP-L6A are consistent, we did not do additional mouse infection studies to measure *speB* transcript levels in infected tissues collected at 72 hpi or later.

5. I would expect that the authors would have considered this point, but was there any attempt

made to undertake the crystallisation study (RopB-CTD + SIP) at lower pH to directly visualize structural changes?

Response: Our efforts to crystallize RopB-CTD-SIP complex in low pH are unsuccessful so far. It is evident that the low pH conditions increase RopB stability in solution (as shown by our thermal stability data). However, we hypothesize that the structural changes introduced in RopB-CTD in low pH is disruptive to crystal packing under the existing crystallization conditions, which prevents the crystallization. Alternatively, the RopB-CTD-SIP complex prepared in low pH may crystallize in a new crystallization condition. This would require *de novo* crystallization screening, obtaining diffraction quality crystals, data collection, structure determination, refinement, and structural analyses. Those experiments are ongoing but such studies are beyond the scope of this study.

6. Bottom of page 15. The authors mention that the Y182 mutation did not result in protein misfolding. How was this accomplished? Was CD spectrometry (or other technique) undertaken to confirm?

Response: As indicated for other recombinant RopB mutant proteins in the earlier sections (page 13, characterization of SIP-contacting amino acid residues in RopB for SIP binding and virulence regulation), we have used 3 criteria to assess proper folding of mutant proteins; WT RopB-like solubility of the overexpressed recombinant protein in the cell lysate as determined by SDS-PAGE analyses, WT RopB-like dimerization as assessed by gel filtration chromatography, and the ability to concentrate the recombinant RopB mutant proteins to concentrations similar to WT RopB. All the RopB mutant proteins in this study had WT RopB-like phenotypes in all 3 categories, indicating that the mutant proteins are not misfolded and form WT-like dimers. We have

presented one representative evidence, the coomassie-stained SDS PAGE gel image, showing equal amount of the purified recombinant WT and mutant RopB mutant proteins in similar volumes in the supplementary figure S8.

7. For virulence experiments (Figure 4E and 6I), can the authors provide information in the Figure legends on the individual dose for each WT and mutant represented in the figure panels?

Response: We have included the dose of the inoculum used in the figure legends.

Finally, the authors work is exciting and novel. This paper provides a detailed model for the regulatory control of an important GAS virulence determinant.

Response: We thank the reviewer for judging our work exciting, novel, and providing a detailed model for the regulation of a major GAS virulence factor.

Mark Walker

Reviewer #2 (Remarks to the Author):

In this manuscript, Do and colleagues explore how pH influences SpeB production via a quorum sensing (QS) peptide (SIP) interaction with the global regulator RopB. Notably, pH has been shown to control other QS pathways in other bacteria but this is the first study to discover specific molecular mechanisms that help explain this phenomenon. This study is detailed and happily, many times I asked an experimental question the authors went on to answer it. The real power in the data is that it includes genetic, phenotypic, biochemical and in vivo demonstrations of pH regulation of SpeB production (via the SIP-RopB pathway). As

mentioned before, there has been no obvious mechanism for pH-dependent control of QS in bacteria and this paper should spur others in the field to examine pH control of QS in other bacteria. My comments on the paper can be found below:

Response: We thank the reviewer for judging our work detailed and multidisciplinary. We do anticipate that this study is likely to provide foundation for similar studies regarding pH control of other QS systems.

In some cases, statistics (as shown with *, etc) are used, and in other cases they are not. Examples where statistics may help follow: Figure 1B, Figure 2A, Figure 3D and 3F. Consistency would help the reader determine what is significant and what is not.

Response: As the reviewer noted, to help the reader to determine statistical significance between samples in the data, we included the results of statistical analyses in all the relevant figure panels.

Figure S4B: The affinity for SIP binding to RopB is obviously increased at lower pH (reduced K_d). The dynamic range (total change in mP) is reduced with pH. Can the authors explain why this has happened?

Response: We believe that pH-dependent photo bleaching of the fluorophore occurs in lower pH conditions, which affects the total change in mP. However, we want to note that the observed total intensities in all pH conditions are consistent among replicates and well above the background (50-fold or higher compared to background).

Minor comments

The line “Each component of the SIP signaling pathway must be functional for a wild-type virulence phenotype” is repeated on page 4.

Response: We have corrected the duplication of the statement in the revised draft.

I appreciate that the authors have explored in great detail pH mediated regulation of SIP-RopB and have an excellent working model (Fig. 7). SpeB production is very complex and there is a limit to how much detail can be provided (a few excellent reviews are cited). However, do the authors believe that pH could influence other areas of the pathway e.g. the endopeptidase PepO, or CovRS?

Response: The influence of pH on the enzymatic activity of PepO and signaling mechanism of CovRS remains a viable possibility. As shown in our studies (the pH preference for SpeB protease activity, Supplementary Figure 3), enzymatic reactions function optimally in a narrow range of pH conditions. Thus, the pH dependence of PepO activity is possible. Similarly, pH-sensing histidine switches have been identified in extracellular sensor kinases similar to CovS (PMID:17466620). Thus, the possibility that CovRS signaling is pH-dependent cannot be ruled out.

Reviewer #3 (Remarks to the Author):

This review covers mainly the structural biology portion of the study. The authors previously determined the crystal structure of apo RopB. Here, they report the structure of RopB bound to its activating peptide SIP. The proposed model of RopB activation is moderately supported by the structural data:

- The proposed acidity-induced conformational change mediated by the histidine switch is not actually observed in this structure.
- The activating peptide seems to be bound to a low-binding-affinity conformation of the protein.

- The molecular mechanism by which SIP binding promotes RopB binding to DNA is still unknown, as the DNA-binding domain of the protein is absent from the structure.

On the other hand, the proposed model is more strongly supported by the accompanying biochemical and biological data.

Response: We thank the reviewer for the detailed comments on the structural biology portion of this study. As the reviewer noted, we have explicitly stated in the manuscript that the structural basis for pH-dependent SIP binding and gene regulation by RopB (i.e. acidic pH-induced allostery mediated by the histidine switch) was not evident from the RopB-CTD-SIP structure (page 15). However, the location of participating amino acids in RopB-CTD structure or RopB-CTD-SIP structure combined with the strong biochemical and biological data strongly support the proposed model for pH-dependent gene regulation by RopB.

We agree with the reviewer that the peptide is likely to be bound to a low affinity state. We have stated that in the revised draft (see our response to reviewer's comment 6, pages 11-12 in the revised draft).

Several issues need to be addressed:

1 - There seems to be a major error in the built structure of the protein (as well as in the authors' previously published structure of apo RopB in the same crystal form, PDB 5DL2). Two chains of RopB are found in the asymmetric unit (ASU), A and B. They are almost completely built but both are missing residue 180. The segments 159-179 and 181-200 form antiparallel helices, and in the current model, residue 180 appears to form a loop between the two helices. This helix-loop-helix segment is located very close to its crystallographic symmetry mate from the adjacent ASU (which also contains two chains of RopB, A' and B'). There is no electron

density at the implied position of residue 180. However, there is clear continuous electron density from the helix 159-179 leading into helix 181'-200' of the adjacent ASU, and vice versa from helix 159'-179' of the adjacent ASU leading into helix 181-200. Thus, residues 159-200 form one long continuous helix, and half of the protein chain was built into the wrong ASU. Attached are pictures of the original and corrected arrangements. One full RopB chain is in green and its crystallographic symmetry mate, in blue. Residues 179 and 181 are indicated by small and large spheres, respectively. This analysis applies to chain B as well. The model should be rearranged to correct this, or an explanation should be provided. This potential error actually has little impact on the structural analysis, but it would imply that the SIP binding site and the histidine switch are composed of residues from both copies of the crystallographic dimer. If the authors do apply this correction, they should also consider correcting their previously reported apo RopB structure.

Response: We are aware of the differences between the final RopB-CTD model and the electron density map after refinement, as identified by the reviewer. Originally, the *de novo* phasing and initial model building was performed using a low resolution (3.8 Å) MAD dataset. As expected for any low-resolution diffraction data set, the electron density for the helices appeared as long tube of densities without clear definition of peptide groups or side chains. Tracing the loops between helices was even more challenging. Specifically, the region corresponding to the helices containing amino acids 176-183 was ambiguous. We were unable to discern the densities for the aromatic side chains (Y176, Y181, and Y182) and the potential loop connecting the two helices. We faced with two possibilities: build the region either as a long helix as the reviewer indicated or as a broken helix as we built. To resolve this ambiguity and overcome the major limitations associated with low resolution diffraction dataset, we took advantage of the availability of high-resolution full-length structures of RopB structural homologs.

We used two well-known structure prediction programs, I-TASSER and Phyre, to identify RopB structural homologs. The following structures have been identified to share high degree of structural similarity with RopB: i) a Rgg regulator from *Listeria monocytogenes* (PDB code: 4RYK, 2.09 Å), ii) Rgg regulator from *S. dysgalactiae* (PDB code: 4YV9, 1.95 Å), iii) PrgX from *Enterococcus faecalis* (PDB code: 2AWI, 2.25 Å), iv) PlcR from *Bacillus thuringiensis* (PDB code: 3U3W, 2.4 Å), v) ComR from *S. thermophilus* and *S. suis*. (PDB code: 5JUB, 2.57 Å; PDB code: 5FD4, 2.9 Å), and vi) ElrR from *E. faecalis* (PDB code: 5V5T, 2.15 Å) also identified to be RopB structural homologs. Similar results were obtained in DALI structural homology search using the RopB-CTD structure as the search model. These results are not surprising as RopB and all the identified RopB structural homologs belong to RRNPP family of quorum sensing transcription regulators.

Structurally, all these regulators have a C-terminal peptide-binding domain with tetratricopeptide repeat motifs (TPR motifs) and a N-terminal DNA-binding domain. Each TPR motif is made up of a pair of antiparallel helices connected by a loop (Fig. 1). Without exception, the C-terminal domains of all the structurally characterized RRNPP regulators have a total of 5 TPR motifs (or 5 pairs of antiparallel helices), which generate a right handed super-helix. The super-helix structure forms an exterior convex surface and an internal concave surface that constitutes the peptide binding site in each subunit. Using these structural observations as a guidance during initial model building, we chose to build the region containing amino acids 159-200 in the current conformation (Fig. 1). Furthermore, building a long helix, as suggested by the refined electron density map, will be disruptive to the folding of typical TPR motifs. Subsequently, we used the relatively better but low-resolution native dataset (3.5 Å for apo structure and 3.3 for SIP-bound structure) for the refinement. Although the refined electron density map indicated

the likelihood of continuous helix, we chose the broken helix conformation due to the low-resolution data set and conformity with the high-resolution structures of the homologs (Fig. 1). In summary, to overcome the model building challenges associated with low-resolution diffraction data, we combined the power of high-resolution full-length crystal structures of RopB structural homologs and x-ray diffraction data to guide our initial model building.

Importantly, as the reviewer noted, the differences in the suggested and our presented build of the model do not affect the mechanistic interpretations or conclusions drawn in this study. Finally, we have included a write up in the methods section acknowledging the possibility of alternative conformation in the electron density and explaining our reasons to build this way (pages 24-25).

Figure 1

Figure 1. Structural overlays of RopB-CTD and the high-resolution full-length structures of RopB homologs. The C-terminal domain of a RopB-CTD subunit is superimposed to the C-terminal domains of PrgX from *Enterococcus faecalis* (a) (PDB code: 2AWI, 2.25 Å resolution), Rgg from *Listeria monocytogenes* (b) (PDB code: 4RYK, 2.09 Å resolution), Rgg from *Streptococcus dysgalactiae* (c) (PDB code: 4YV9, 1.95 Å resolution), and ErIR from *E. faecalis* (PDB code: 5V5T, 2.15 Å resolution). Ribbon representation of individual subunits are shown. The C-terminal domain of RopB-CTD is colored in light grey, whereas RopB structural homologs are shaded in dark grey. The region corresponding to RopB amino acids 159-200 shown in the boxed region in the left panel is magnified in the inset on the right panel. The amino acids connecting the helices $\alpha 7$ and $\alpha 8$ of TPR motif 3 in RopB-CTD are colored in purple, whereas analogous region in the structural

homologs of RopB are colored in red. Similarly, the loop connecting the helix 1 and helix 2 of TPR motif 3 was clearly visible in the structures of PlcR from *Bacillus thuringiensis* (PDB code: 3U3W, 2.4 Å resolution), and ComR from *S. thermophilus* and *S. suis*. (PDB code: 5JUB, 2.57 Å; PDB code: 5FD4, 2.9 Å resolution) (data not shown).

2 - There are a few Ramachandran outliers or borderline values in the backbone angles of the SIP structure. Notably, a cis peptide bond was modeled at position 6-7. As the electron density for the peptide is not clearly defined at this resolution, this should very likely be corrected to a trans bond.

Response: Corrected. Please see the attached new validation report from PDB.

3 - Figure 3A:

Do the authors mean “composite omit electron density map”? Could they verify that this is indeed a 2fo-fc map displayed at the 1-sigma level? The map file that they provided, although being a refinement output map and not a composite omit one, shows weak fragmented density for the peptide at the 1-sigma level, and only shows a similar density to Figure 3A when contoured at 0.5-sigma.

Response: The presented figure is indeed a 2Fo-Fc map displayed at 1-sigma level. We have made the figure using Pymol. We have provided the image from our analyses below (Fig. 2).

Figure 2

4 - Section Structural Basis of SIP Recognition by RopB:

- “stabilizing polar contacts between the side chains of N152 of helix a6, N192 of helix a8, and K278 of helix a12 of RopB and the peptide backbone of SIP”

The side chains of N152 and K278 are quite far from the peptide backbone, as seen in Figure 3B and in the coordinates file provided. Rather, the hydrogen bonds between RopB and SIP involve the side-chains of Y224 and N192, as listed correctly in Figure S6A. All the residues

mentioned throughout this paragraph should be double-checked and Figure 3B modified accordingly.

- “hydrophobic contacts between the side chains of SIP amino acids and the side chains of F155 of helix a6, I195 of helix a8, and M267, F268, and Y271 of helix a12” V191 also establishes hydrophobic contacts with the peptide.

Response: The details of contacting residues have been double checked and the text has been revised accordingly.

5 - Figures 4 and S7, and the related sections in the Results:

One of the mutants analyzed is N152A. However as mentioned in #4, N152 is quite far from the peptide (over 5 Angstroms from the L5 side chain and even further from the SIP backbone, as seen in Figure 3B and in the coordinates file provided). The authors should explain this or remove this mutant from the Results.

Response: We agree with the reviewer’s observation that the side chain of N152 is too far away to interact with SIP in the structure. In accordance with this, we revised the section indicating that N152 is not involved in direct interactions with SIP. We have provided our rationale for including N152 in the functional analyses (page 13).

6 - Section Structural Basis of SIP Recognition by RopB:

“However, no direct interaction was observed between the side chain of L6 of SIP (SIP-L6) and RopB. Together, these data indicate that each amino acid of SIP except the side chain of L6 is required for sequence-specific SIP recognition by RopB.” Residues 1-4 of SIP establish few contacts with RopB, and those are weak non-specific Van der Waals interactions between hydrophobic side chains of the peptide and polar or charged residues of the protein. The structure alone does not seem to explain the importance of this

portion of SIP for its recognition by RopB, although the biochemical and biological data do support the importance of all these residues. The binding affinity of SIP to RopB is around 10nM, and although the peptide was present at relatively high concentration during crystallization, its average crystallographic B-factor is two times higher than the protein's, indicating conformation heterogeneity or partial occupancy. Also, SIP is absent from the second RopB copy in the asymmetric unit. The authors propose a pH-induced conformational change in the protein that increases its affinity for the peptide. Based on all these points, have they considered the possibility that the current structure (at neutral pH) may represent a low-binding-affinity conformation in which RopB can still associate with SIP but without forming many of the specific interactions implied by their biochemical data that would result in nanomolar affinity? The authors should discuss this point if they deem it relevant.

Response: We agree with the reviewer's comment that the RopB-CTD-SIP structure is likely to represent a low affinity conformation. The following factors may contribute to the low affinity state: i) the full-length RopB binds to SIP with a K_d of approximately 10 nM. But the RopB-CTD used in the crystallization studies engages in a relatively low affinity interaction with SIP (K_d 44 nM, Supplementary Fig. 5) compared to full length RopB. ii) We were able to see the electron density corresponding to SIP in only one of the two RopB-CTD subunits in the asymmetric unit, indicating that crystal packing also interfere with the access of peptide-binding pocket of RopB-CTD to SIP. iii) Crystallization was done in near neutral buffer pH condition (pH 7.5) that is not ideal for SIP binding (pH 7.5, K_d of 42 nM, 23-fold reduction in affinity, Fig. 2d). Our efforts to crystallize RopB-CTD-SIP complex in low buffer pH under the current crystallization condition is futile so far, suggesting that RopB-CTD-SIP may exist in a different conformation in low pH that is disruptive to the crystal packing in the current crystallization condition. Due to these above-mentioned reasons, we acknowledge that

the RopB-CTD-SIP structure presented in this study may represent a low affinity state. Nevertheless, we note that the electron density for SIP is clearly discernible in the structure and importantly, the RopB-SIP interactions identified in the structure are fully validated by genetic, biochemical analyses, and mouse infection studies. However, in addition to the RopB-SIP interactions detailed in this study, we anticipate that additional interactions may occur between RopB and SIP in low pH that contributes to high affinity interactions. We have discussed these analyses briefly in the revised draft (pages 11 - 12).

7 - Related to #6, there should be a figure, at least in the supplementary, comparing the overall structure (for example in ribbon form) of the SIP-bound chain A crystallographic dimer with that of chain B and with the dimers from the previously published apo RopB structure. This would illustrate that the conformations of apo RopB and SIP-bound RopB from this neutral-pH structure are almost identical.

Response: As the reviewer suggested, we have included a supplementary figure 6 illustrating the following: i) superposition of SIP-bound RopB-CTD chain A to SIP-bound RopB-CTD chain B, and ii) superposition of apo RopB-CTD and SIP-bound RopB-CTD.

8 - Have the authors considered a potential role for residue E185 in the histidine switch? It is located close to H144, is highly conserved and could form a salt bridge with the latter at moderately acidic pH. The pH dependency of the interaction between H144 with Y182 is unclear, as tyrosine is uncharged and can act both as hydrogen bond donor and acceptor, and can participate in aromatic stacking. The data does implicate Y182, but there may be a larger conformational change at acidic pH, than only rotation of Y182 towards H144 or vice versa. In addition, if the authors implement the correction suggested in #1, this would mean that H144

interacts with a residue (Y182 or maybe E185) from the second subunit of the dimer, with possible implications in dimer formation or stability for the histidine switch.

Response: We anticipated that additional interactions between the protonated side chain of H144 and other yet-to-be identified amino acid residues occur at the base of the SIP-binding pocket (in addition to the interactions characterized). However, we have picked few representative residues for the mutational analyses. Nevertheless, we do agree with the reviewer's analysis that the location and chemical properties of E185 is ideal for its interactions with the protonated side chain of H144 in low pH. Thus, in accordance with the reviewer's comment, we introduced alanine substitution at E185 in GAS genome and characterized the isogenic *ropB-E185A* mutant strain for *speB* expression by qRT-PCR. Consistent with the structural prediction, the E185A mutant strain had decreased *speB* expression, secreted SpeB levels, and SpeB protease activity (Fig. 6a-d). The transcript levels of *speB* in the isogenic E185A mutant strain was comparable to that of Δ *ropB* mutant strain (Fig. 6c). We have also purified recombinant E185A mutant protein and characterized the mutant protein for SIP binding. No alterations in the dimerization properties of E185A was observed, suggesting that E185 does not participate in RopB dimerization. The affinity of the E185A mutant protein for SIP was drastically decreased, indicating that the side chain of E185 is critical for SIP binding by RopB (Fig. 6b). The data is included in the revised figure 6. The role of E185 in the putative hydrogen-bonding network with the protonated side chain of H144 has also been incorporated in the model figure (Fig. 7).

9 - As the main novelty of the structure is the complex between RopB and SIP, it could be interesting to compare it to related proteins. For example, the TPR-containing, quorum-sensing regulator PlcR also undergoes a conformational change upon binding to a signaling peptide

(PMID 23277548), albeit not pH-induced. Such a comparison could give clues as to how SIP binding to RopB stimulates DNA binding by the protein.

Response: As the reviewer noted, the structures of PlcR-PapR-DNA ternary complex and PrgX-peptide binary complex are available. Comparison of protein-peptide and protein-peptide-DNA complexes revealed the peptide-induced/DNA-induced allosteric changes occurring in the DNA-binding domain of PlcR that modulate its DNA binding. Similarly, comparison of the structures of apo PrgX and PrgX-peptide complexes identified the structural changes occurring in the C-terminal tail of PrgX and elucidated the likely mechanism of allostery. We have reviewed this topic recently (PMCID: PMC4938729). A key difference is that the structure reported here is SIP bound to the C-terminal domain of RopB that lacks the entire DNA binding domain (amino acid residues 1-56), not the full-length RopB. Furthermore, RopB also does not have the C-terminal tail similar to PrgX.

The RopB-CTD-SIP structure is highly informative as it provides the structural basis for SIP recognition by RopB. Previously, we have demonstrated that the DNA-binding domain of RopB is required for SIP-mediated oligomerization of RopB on DNA (PMCID: PMC5635878). Thus, the structural overlays of RopB-CTD without the DNA-binding domain onto its structural homologs are unlikely to yield information on the peptide-induced allostery occurring in the DNA-binding domain (or elsewhere). Consequently, such analyses are unlikely to be informative regarding how SIP binding to RopB stimulates DNA binding by the full-length protein. Consistent with this, our structural overlay analyses in response to reviewer's comment 1 failed to reveal the mechanistic details on the SIP-induced allostery in RopB.

10 - Minor points:

- In the third paragraph of the introduction, this sentence is duplicated: “Each component of the SIP signaling pathway must be functional for a wild-type virulence phenotype”

Response: We have corrected the duplication of the statement in the revised draft.

- The Materials and Methods section is not very informative and could be re-written. Instead of listing every type of experiment followed by “details are provided in the supplementary”, the authors should mention the key points here, and have one introductory sentence referring to the supplementary for the detailed Materials and Methods.

Response: In accordance with the reviewer’s comment, we have moved the Materials and Methods from the supplementary section to the main draft of the manuscript.

- Figure 3B: the gray side chains should have their oxygen and nitrogen atoms colored in red and blue, like the other residues in this panel.

Response: Done.

- Figure 5A: the line connecting the base of the SIP-binding pocket of the two subunits does not stand out against the background of the rest of the image. The base of the pocket could be marked in some other way. Also, the residues forming the histidine switch, displayed as all-atom spheres, are difficult to distinguish from each other. They could be shown as sticks, or maybe one sphere per residue.

Response: In accordance with the reviewer’s comment, we have highlighted the line connecting the base of the SIP-binding pocket of the two subunits in bright green to improve its visibility.

Our objective for the panel 5A is to demonstrate the possibility for extensive intra and inter-subunit interactions occurring at the base of SIP-binding pocket. However, in light of the reviewer's comment, we have reduced the size of the spheres to enable the identification of individual amino acids. Furthermore, a close up look of the residues participating in the histidine switch as sticks are shown in panel 5B.

- Figure S8A: the source of the full-length RopB model is not indicated. Is it from a previous publication? If not, details should be provided in the supplementary methods.

Response: We have included the information in the figure legend.

- Table S3: the Wilson B-factor for the dataset should be provided in the Data Collection section.

Response: Included in the revised data collection section.

- At several places in the manuscript, the authors refer to the pH-induced conformational change in RopB. As this rearrangement was not actually observed in the present structure, it should be specified as putative / proposed / probable / likely...

Response: Done.

Alexei Gorelik, McGill University

Reviewers' comments:

Reviewer #3 (Remarks to the Author):

The authors satisfactorily addressed all of my concerns, except for point #1:

RopB crystallizes as a dimer, and the authors built the structure in accordance with related proteins: the segments 159-179 and 181-200 form antiparallel helices, and the residue 180, implied to form a short connecting loop, is missing from the model. This region is in close proximity with its copy from the second subunit of the dimer. I previously suggested that, based on the electron density map, residues 159-200 form one continuous helix, and half of the protein should thus be rearranged into the adjacent asymmetric unit.

The authors responded with two points supporting their original model: the initial apo RopB structure from their previous publication was difficult to build due to its low resolution, and their model agrees with the high-resolution structures of homologous proteins from the RRNPP family. However, the current structure has a better resolution than the initial apo RopB. To verify the arrangement in that region of the dimer, I removed from the model the residues around position 180 (177-179 and 181-183), carried out refinement with simulated annealing and generated a 2Fo-Fc map at the 1-sigma level (attached figure, density displayed around residues 172-188). This map is therefore not biased by the initial model. It clearly shows a continuous helix (and its symmetry-related copy); there is no density at the implied position of residue 180 in the middle, that would bridge the helices. The data thus strongly supports the corrected model that I proposed.

This arrangement would indeed be different from the structures of homologous proteins. RopB is part of a family whose fold is comprised of TPR motifs (helix-loop-helix repeats). TPR-containing proteins from unrelated families sometimes also form dimers, and in several cases, this occurs by "domain swapping" – meaning that a portion of a monomer crosses over into the second monomer. The overall structure of one subunit then looks like a normal monomer, but is actually composed of regions from two protein chains. Here are examples of TPR proteins dimerizing by domain swapping: PMID 25882846, 11377203, 22825553. We have also encountered this in a project in our lab. Sometimes, within a family of TPR proteins, some homologs dimerize "normally", whereas others do so by domain swapping. The corrected structure of RopB appears to be a case of domain swapping.

In summary, I recommend that the structure should be corrected before publication, or to have an additional structural biologist review and adjudicate this point. My apologies for potentially delaying the publication of this interesting study.

Alexei Gorelik, McGill University

Response to Reviewer's Comments:

Reviewer #3 (Remarks to the Author):

The authors satisfactorily addressed all of my concerns, except for point #1: RopB crystallizes as a dimer, and the authors built the structure in accordance with related proteins: the segments 159-179 and 181-200 form antiparallel helices, and the residue 180, implied to form a short connecting loop, is missing from the model. This region is in close proximity with its copy from the second subunit of the dimer. I previously suggested that, based on the electron density map, residues 159-200 form one continuous helix, and half of the protein should thus be rearranged into the adjacent asymmetric unit. The authors responded with two points supporting their original model: the initial apo RopB structure from their previous publication was difficult to build due to its low resolution, and their model agrees with the high-resolution structures of homologous proteins from the RRNPP family.

However, the current structure has a better resolution than the initial apo RopB. To verify the arrangement in that region of the dimer, I removed from the model the residues around position 180 (177-179 and 181-183), carried out refinement with simulated annealing and generated a 2Fo-Fc map at the 1-sigma level (attached figure, density displayed around residues 172-188). This map is therefore not biased by the initial model. It clearly shows a continuous helix (and its symmetry-related copy); there is no density at the implied position of residue 180 in the middle, that would bridge the helices. The data thus strongly supports the corrected model that I proposed. This arrangement would indeed be different from the structures of homologous proteins. RopB is part of a family whose fold is comprised of TPR motifs (helix-loop-helix repeats). TPR-containing proteins from unrelated families sometimes also form dimers, and in several cases, this occurs by “domain swapping” – meaning that a portion of a monomer crosses over into the second monomer. The overall structure of one subunit then looks like a normal monomer, but is actually composed of regions from two protein

chains. Here are examples of TPR proteins dimerizing by domain swapping: PMID 25882846, 11377203, 22825553. We have also encountered this in a project in our lab. Sometimes, within a family of TPR proteins, some homologs dimerize “normally”, whereas others do so by domain swapping. The corrected structure of RopB appears to be a case of domain swapping.

In summary, I recommend that the structure should be corrected before publication, or to have an additional structural biologist review and adjudicate this point. My apologies for potentially delaying the publication of this interesting study.

Response: As we noted in our earlier response, we agree with the reviewer about the possibility of an alternative RopB-CTD conformation in which the region corresponding to 159-200 form a long helix rather than the helix-loop-helix. We discussed our reasoning behind the helix-loop-helix built due to the low-resolution nature of diffraction data and its conformity with the high-resolution structures of RopB structural homologs. Compared to our apo RopB-CTD diffraction data, the current data set is higher resolution (3.1 Å compared to 3.8). The diffraction data from RopB-CTD-SIP crystals strongly suggests (as noted by the reviewer) the presence of single long helix in the region corresponding to amino acids 159-200. Thus, in accordance with the reviewer’s suggestion, we have rebuilt that region as single long helix and revised the figures accordingly (Fig. 3a, Fig. 5a, Fig. 5b, and Fig. 7). We have included a supplementary figure showing the electron density corresponding to single long helix and a side by side representation of previous and revised RopB-CTD models. With the exception of domain swapping, no other differences or changes were observed between the previous and current RopB-CTD-SIP model. We have submitted the revised PDB file to the database under the same code.

We would like to emphasize that the differences in the old and revised build of the RopB-CTD-SIP model do not affect the mechanistic interpretations or conclusions drawn in this study or our previous study. Finally, we have included a write up in the results and methods section explaining the differences in the conformation of RopB-CTD in the region corresponding to amino acids 159-200 compared to our previously published model (pages 24-25).

REVIEWERS' COMMENTS:

Reviewer #3 (Remarks to the Author):

The authors satisfactorily addressed all of my concerns.